# A study on multi-objective optimization for the location selection of smart underground parking facilities in high-density urban areas of megacities: A case study of Jing'an district, Shanghai

**Xiaodan Li[1,2], Yunci Guo[1,2]\*, Zhen Liu[1,2], Dandan Sun[3], Yidi Liu[2], Wencan Wang[3]**

**1** State Key Laboratory for Tunnel Engineering, Beijing, China, **2** China University of Mining and Technology, Beijing, China, **3** China RAILWAY 15 BUREAU Group Corporation, Shanghai, China

\* ZQT2300605122A@student.cumtb.edu.cn

## Abstract

The acceleration of global urbanization and the rapid growth of urban populations have intensified the complexity and urgency of parking demand. In megacities with limited land resources, efficiently addressing diverse parking needs has become a critical issue for sustainable urban development. Multi-objective optimization methods are widely applied to tackle such challenges, providing decision-makers with a set of optimal solutions that balance multiple objectives. However, existing studies often lack quantitative analyses of the relationships among these solutions, limiting their applicability in accommodating decision-makers with varying preferences. This study focuses on Jing'an District in Shanghai, a representative region of a Chinese megacity, to address this global issue. Based on real-world data, a multi-objective optimization model is constructed considering convenience, coverage, and cost-efficiency. The model is solved using an improved Non-dominated Sorting Genetic Algorithm II (NSGA-II), which dynamically adjusts crossover and mutation rates. Furthermore, the Pareto solution set is quantitatively analyzed from a cost-benefit perspective by integrating marginal benefit theory. This approach provides robust support for decision-makers seeking an optimal balance between cost and benefit, offering scenario-specific strategies. The findings of this study not only present an innovative, systematic, and flexible solution to the "parking dilemma" in high-density residential areas but also provide practical guidance and insights for other large cities in the planning and implementation of smart underground parking facilities.

## 1. Introduction

Globally, the number of private vehicles is steadily increasing due to advancements in technology, economic growth, and improved living standards. This trend has

**Data availability statement:** All relevant data are within the manuscript and its Supporting Information files.

**Funding:** The author(s) received no specific funding for this work.

**Competing interests:** The authors have declared that no competing interests exist.

exacerbated the conflict between limited land resources and the rising number of motor vehicles in major cities across North America, Europe, and East Asia [1–4]. Particularly in the high-density districts of megacities, the tension is amplified by constrained land resources and diverse parking demands from residents [5], making "parking difficulties" a ubiquitous challenge. Concurrently, the continuous scarcity of land resources in urban centers has rendered traditional large-scale surface parking facilities problematic. These facilities not only occupy substantial land and are challenging to locate but also contribute to environmental and economic issues, such as exacerbation of the urban heat island effect, degradation of cityscapes, and increased infrastructure maintenance costs [6]. Addressing the urgent need to efficiently provide sufficient parking spaces within limited spaces has become a central concern in urban management and transportation planning.

According to a public report from the Traffic Management Bureau of China's Ministry of Public Security, the number of motor vehicles in China surged from 264 million in 2014–453 million in 2024 (https://www.mps.gov.cn/; accessed on 19 February 2024). In response to the structural conflict between urban land shortages and surging parking demands, various regions are actively exploring the integration of advanced technologies with underground spaces to construct smart parking facilities for optimal land utilization. Shanghai, as a representative megacity with high-density urban areas in China, has promulgated relevant policy documents to alleviate the contradiction between the soaring number of vehicles and land scarcity. For instance, the "14th Five-Year Plan for the Development of Shanghai's Parking Industry" explicitly emphasizes the need for efficient use of land resources, rational utilization of aboveground and underground spaces, and expedited planning and construction of public parking facilities. Specific recommendations include innovating parking solutions for high-demand areas such as old residential neighborhoods, hospitals, and schools to optimize parking resource allocation and enhance urban traffic efficiency (https://www.shanghai.gov.cn/; accessed on 20 February 2024).

Aligned with this policy direction, in 2023, China Railway 15th Bureau completed the construction of the world's largest-diameter and Shanghai's first EUP vertical tunneling (shield) underground smart parking facility in Jing'an District, Shanghai. By employing vertical excavation and modular design in urban "corner spaces," this facility significantly reduces surface environmental encroachment and provides substantial additional parking capacity for old communities [7]. This underground approach demonstrates considerable advantages in land utilization over traditional facilities and aligns, to some extent, with the demands for sustainable urban development. However, converting technological advantages into effective parking site selection plans requires balancing multiple objectives and maintaining flexibility amid dynamic demands and policy environments. This multi-objective decision-making process with varying constraints imposes higher-level challenges on urban managers and planners [8].

To transform technological advantages into actionable parking site selection solutions, it is essential to balance conflicting objectives such as accessibility, population coverage, construction costs, and environmental impacts [9], while

maintaining adaptability to dynamic demands and policy shifts. Existing research on parking site optimization often employs Geographic Information Systems (GIS) for spatial analysis and visualization [10] and utilizes Multi-Criteria Decision Analysis (MCDA) or Multi-Objective Decision-Making (MODM) methods to balance diverse objectives [11]. MCDA methods commonly assign weights to evaluation criteria using approaches like Analytic Hierarchy Process (AHP) or fuzzy evaluation, balancing different decision criteria. However, these methods are significantly influenced by weight settings, which, if unreasonable or overly subjective, may affect the final decision outcomes [12]. By contrast, MODM methods generate a Pareto-optimal set under multiple conflicting objective functions, offering decision-makers a broader range of feasible solutions and a clearer trade-off process [13]. Yet, most studies remain at the stage of merely enumerating Pareto solutions, lacking in-depth exploration of quantified differences among solutions in the set. This deficiency hinders the ability of decision-makers with differentiated needs or preferences to finely weigh costs, benefits, and application scenarios [14]. Furthermore, in dynamic urban environments characterized by evolving parking demands and policy orientations, traditional static optimization approaches often fail to adapt promptly. In this study, a multi-objective optimization model was constructed based on three key dimensions: service convenience, resource coverage efficiency, and economic feasibility. Among these, walking distance serves as a critical indicator of the attractiveness and usability of parking facilities, directly reflecting the travel cost incurred by pedestrians from the nearest entrance to their destination. Additionally, population coverage rate is employed as a core metric for evaluating the service capacity of parking facilities, quantifying the extent to which parking infrastructure meets demand across different regions. Lastly, construction cost is identified as a decisive factor in investment returns and implementation feasibility, directly influencing the economic viability and scalability of location selection under budget constraints. These three objectives collectively encompass user experience, public service effectiveness, and economic return on investment. Moreover, they counterbalance one another, enabling decision-makers to formulate integrated location-selection strategies that accommodate dynamic demand and budget limitations while balancing convenience, coverage, and cost efficiency.

In summary, this study constructs a multi-objective optimization model and utilizes the improved NSGA-II algorithm to obtain a Pareto-optimal set. Subsequently, marginal benefit theory is introduced during the post-Pareto analysis phase to conduct incremental benefit analyses for each candidate solution. By focusing on the changing trends of solutions in the "cost-benefit" dimension, this study aims to assist managers in intuitively assessing the potential value and trade-offs among solutions. The key contributions of this research are given below:

(1) By improving the NSGA-II algorithm and incorporating a dynamic adjustment strategy for crossover and mutation rates, this study enhances the convergence speed of NSGA-II while maintaining population diversity. These improvements provide a more stable and efficient method for addressing the complex problem of parking facility location selection.

(2) For the first time, marginal benefit analysis is integrated into post-Pareto analysis, enabling a quantitative examination of the relationships between various solutions. This approach yields dynamic decision-making strategies tailored to different scenarios. The proposed strategy adapts to changes in urban demands and policies over time, allowing for the selection of optimal location solutions that balance sustainable urban development with flexible governance needs. It provides decision-makers with diverse, scenario-based decision support.

(3) Using real spatiotemporal data from Jing'an District, Shanghai, this study focuses on the development of smart parking facilities. By effectively utilizing urban fringe spaces, the model balances convenience, coverage, and investment costs, offering a replicable paradigm for addressing the parking challenges of high-density urban areas in megacities.

The remainder of this paper proceeds as follows. Section 2 is the literature review. Section 3 introduces the research methodology and data. Section 4 provides empirical analyses. The final section draws the discussions and conclusions.

## 2. Literature review

In the fields of urban transportation planning and land use, site selection has consistently been a pivotal research topic. A range of technical approaches, including Geographic Information Systems (GIS), Multi-Criteria Decision Analysis (MCDA), and Multi-Objective Decision Making (MODM), have been extensively employed in existing studies. In architecture and planning, GIS technology, known for its robust spatial analysis and data visualization capabilities, has long been utilized in site selection and planning research [15]. Tang et al. [16], for example, employed the spatial analysis functionality of GIS by integrating Point of Interest (POI) data with machine learning techniques, using decision tree models to optimize parking lot site selection. Similarly, Itzhak Benenson et al. [17] proposed the PARKAGENT model, combining GIS data with field surveys to provide empirical support for the formulation of parking policies. These studies highlight the advantages of GIS in the early-stage feasibility analysis and geographic data processing of site selection; however, when it comes to comprehensive trade-offs among multiple objectives, relying solely on GIS capabilities proves insufficient. Thus, integration with other decision-making methods is often required [18].

MCDA focuses on providing comprehensive evaluation results for decision-makers across multiple criteria dimensions. The Analytic Hierarchy Process (AHP), introduced by Saaty [19] in 1980, has been widely applied in MCDA issues, including site selection for wind farms [20], emergency shelters [21], solar power stations [22], and landfills [23]. Many studies have also applied AHP in combination with GIS to address parking lot site selection [24–26]. For instance, Kazazi Darani et al. [27] integrated AHP with the Technique for Order of Preference by Similarity to Ideal Solution (TOPSIS) to identify optimal public parking lot locations in Boroujerd, Iran. Demir et al. [28] adopted a GIS-based fuzzy AHP approach to evaluate parking lot sites in four districts of Istanbul. This GIS-based fuzzy AHP method has also been applied to parking management in Shiraz's central business district in Iran, offering policymakers insights into site selection and investment considerations [29]. Beyond AHP, methods such as Ordered Weighted Averaging (OWA) have been employed in parking lot site selection studies [30–32]. Jelokhani-Niaraki et al. [33] integrated OWA with GIS into a web-based platform, proposing an effective Multicriteria Spatial Decision Support System (MC-SDSS).

Parking lot site selection often involves multiple conflicting objectives, such as cost, accessibility, and environmental impact [34]. MODM is more suitable for scenarios requiring the balancing of such conflicting objectives [35,36]. By constructing mathematical models that simultaneously consider multiple objective functions, MODM generates a Pareto-optimal set, offering decision-makers flexible options based on specific needs.

In recent years, multi-objective optimization algorithms, along with many-objective (MaO) techniques designed for higher-dimensional problems, have undergone rapid advancements. These developments have led to a range of refined optimization models, providing extensive technical support for location selection research involving multiple conflicting objectives in complex environments. A detailed comparison of the advantages and limitations of nine commonly used multi-objective optimization algorithms is presented in Table 1. Given that this study focuses on three objective functions and incorporates economic benefit quantification and dynamic strategy classification in the post-Pareto analysis phase, the NSGA-II algorithm [46] was prioritized for its maturity in engineering applications, strong interpretability, and balanced trade-off between computational efficiency and solution quality. To further enhance its performance, dynamic crossover and mutation rate adjustments were introduced. Current research on the application of NSGA-II has yielded significant advancements. For instance, Shen et al. [47] employed genetic algorithms combined with a dynamic Thiessen polygon partitioning method to construct an optimized public parking facility location model. This approach not only improved computational complexity compared to earlier methods but also enhanced Pareto front distribution through mechanisms such as fast non-dominated sorting and crowding distance calculations. Souier et al. [48] leveraged NSGA-II to provide dynamic decision support for flexible manufacturing systems. Hong et al. [49] applied NSGA-II to optimize the location of electric vehicle charging stations, considering multiple objectives such as construction cost, demand, and coverage. These studies demonstrate the effectiveness and adaptability of NSGA-II in multi-objective optimization scenarios, highlighting its strengths in balancing conflicting objectives and generating high-quality Pareto solution sets [50r].

**Table 1. Comparative overview of multi-objective optimization algorithms.**

| Algorithm | Advantages | Limitations |
|---|---|---|
| MOCLO [37,38] | Superior WSN coverage and energy efficiency; Strong robustness. | Primarily designed for WSN scenarios; Performance in other domains remains unverified. |
| MORKO [39] | Fast convergence; Significantly outperforms various MOEAs. | Slight increase in computational complexity; May converge prematurely; High sensitivity to parameter settings. |
| MODHHO [40] | Optimal AUC and TPR for IoT Botnet detection across five datasets; Lower computational overhead compared to similar multi-objective approaches. | Limited to discrete problem evaluations; Generalizability needs further verification. |
| MOMRFO [41] | Integrates Manta Ray Foraging with external elite archives; Adaptive archive mechanism enhances convergence and diversity. | Encounters challenges when handling discontinuous Pareto fronts. |
| MOEDO [42] | Balanced IGD/HV performance with statistically significant improvements. | Lacks optimization capabilities for discrete and binary problems; Computational complexity increase remains minimal. |
| MaOGOA [43] | Demonstrates superior performance in high-dimensional spaces; Enhances convergence speed and diversity. | Efficiency and resource consumption may become limiting factors in ultra-high-dimensional optimization scenarios requiring large population diversity. |
| MaOMVO [44] | Introduces an innovative information feedback mechanism for population updates; Exhibits strong stability across various optimization tasks. | Limited adaptability to degenerate Pareto fronts. |
| MaOWOA [45] | Performs exceptionally well in LSMOP high-dimensional problems. | Complex algorithmic process; Computational cost is higher than the original WOA. |
| NSGA-II [46] | Well-established and widely applied in engineering; High interpretability; Balances computational efficiency and solution quality. | Still possesses relatively high complexity; Less effective in highly dimensional scenarios. |

Nevertheless, most existing studies focus on "how to obtain the Pareto set" but lack in-depth analysis of the quantitative differences among solutions within the set [51]. When decision-makers face large-scale Pareto sets, the absence of systematic evaluations of incremental benefits and potential trade-offs among solutions often hampers the precise selection of the most suitable option for specific scenarios. In practical applications, some researchers have attempted to use MCDA for post-Pareto set filtering to select compromise solutions [52–60], with TOPSIS being the most commonly applied method [61]. Others have employed methods such as fuzzy-based mechanisms and entropy weight techniques. Although such post-Pareto analyses assist in reducing the solution set, they rarely examine the dynamic changes in incremental benefits among objectives within the set. This limitation hinders the effective identification of relative benefit differences among solutions, particularly in complex environments like parking lot site selection, which involve multiple objectives and constraints. Consequently, it becomes challenging to provide decision-makers with enough flexibility and granularity to cater to varying preferences or resource conditions.

To address this research gap, this study introduces marginal benefit theory to quantify the incremental benefits of candidate solutions in key objectives after obtaining the Pareto set through MODM methods. By employing this approach, the study delves deeper into the differences among solutions within the set and assists decision-makers in identifying critical thresholds of diminishing returns in costs and benefits. This method establishes more appropriate site selection strategies, offering dynamic and flexible decision support for the planning of underground smart parking facilities in high-density urban environments.

## 3. Methodology

### 3.1 Study area

This study focuses on Jing'an District, Shanghai (Fig 1), as a case study for site selection optimization. According to the "2023 White Paper on the Development of China's Parking Industry," significant spatial and temporal imbalances exist in the supply and demand for parking resources in Shanghai, particularly in areas near hospitals and

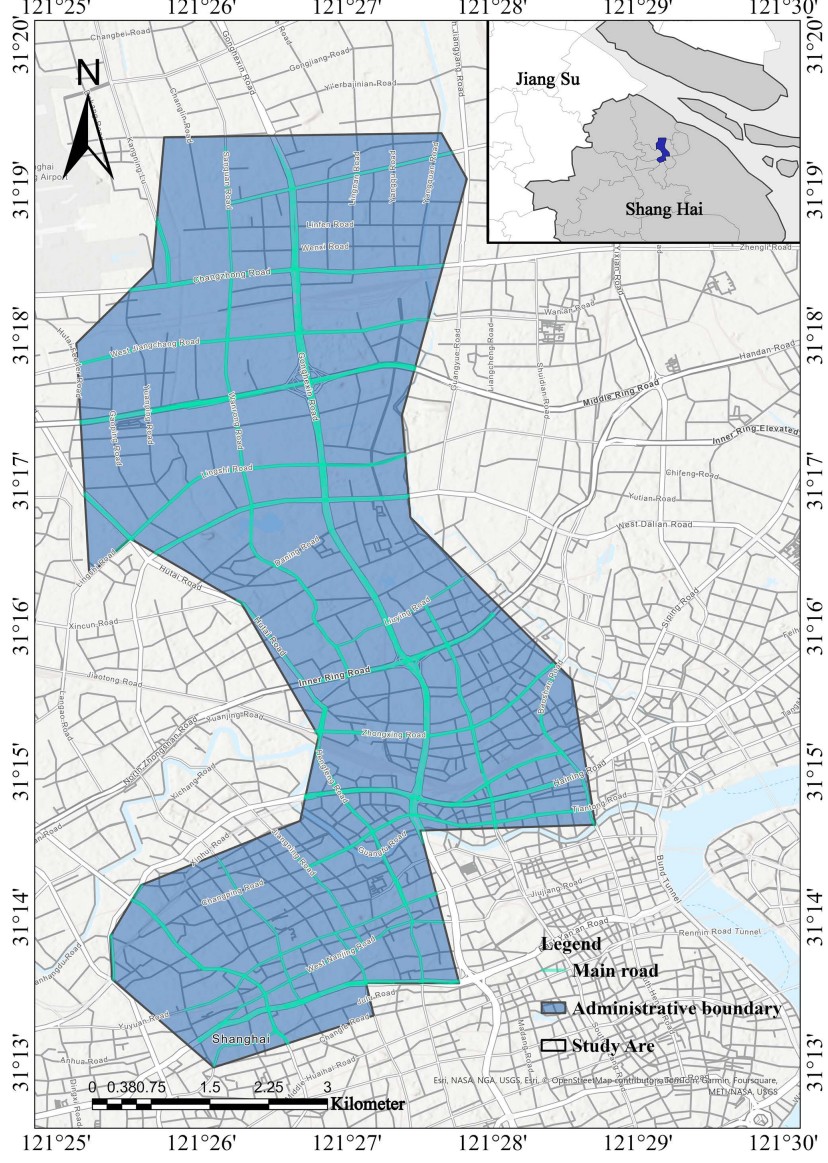

**Fig 1. Overview map of the study area.**

residential neighborhoods (https://www.199it.com/; accessed on 25 February 2024). Additionally, the "Special Plan for Urban Land Space Ecological Restoration in Shanghai (2021–2035)" emphasizes the importance of enhancing urban space ecological restoration and functional optimization to improve the efficiency of intra-urban space utilization. This strategy aims to better address the conflicts between land scarcity and increasing parking demands in high-density environments.

For instance, residential neighborhoods in Shanghai collectively provide approximately 4 million marked parking spaces. However, the number of vehicles parked in residential areas at night significantly exceeds this capacity, with nearly 12% of vehicles occupying unmarked spaces. The proportion is even higher for older neighborhoods, where unmarked parking accounts for approximately 24% (https://jtw.sh.gov.cn/; accessed on 27 February 2024).

## 3.2 Data collection and processing

The data utilized in this study encompasses actual construction data of smart parking facilities, as well as spatial analysis-related datasets, including Point of Interest (POI) and Area of Interest (AOI) data, population data, and transportation network data. After undergoing careful screening and cleaning, these datasets are employed for the construction of models and optimization analyses. The types of data utilized in this study are summarized in Table 2.

## 3.3 Research methodology

This study proposes an integrated decision-making methodology encompassing three stages: "preliminary screening—multi-objective optimization—post-Pareto analysis" (as illustrated in Fig 2). Firstly, in the preliminary screening phase, the potential parking site locations are focused on urban marginal spaces—such as green spaces, parks, underpasses, urban squares, and vacant lands—based on Points of Interest (POI), Administrative Boundaries (AOI), statistical data, and grid data, as well as the specific conditions of the study area. This approach aims to enhance land use efficiency while minimizing the impact on surrounding environments. Secondly, a multi-objective optimization model is constructed, incorporating three objective functions: maximizing population coverage, minimizing walking distance, and minimizing construction costs. The improved NSGA-II algorithm is employed to solve the model, generating a three-dimensional Pareto-optimal solution set. Finally, in the post-Pareto analysis phase, this study introduces marginal benefit theory to quantitatively evaluate trade-offs among objectives from an economic benefit perspective. This process further filters and identifies relatively optimal solutions that demonstrate comprehensive value under varying demand scenarios, providing targeted and flexible references for the site selection of smart parking facilities in high-density urban environments.

**3.3.1 GIS-based spatial analysis.** This study employs ArcGIS to analyze the current spatial characteristics of Jing'an District, focusing on visualizing population distribution and the transportation network (Fig 3). Kernel density analysis is conducted for key areas such as residential neighborhoods, hospitals, schools, and bus stations [62], aiming to reveal the spatial distribution patterns of these areas and the potential parking pressures they may generate. These insights provide a scientific basis for subsequent parking site selection.

Subsequently, urban marginal spaces—such as green spaces, parks, areas under elevated highways, urban squares, and vacant lands—are identified through preliminary site selection screening. Finally, the network analysis tool in ArcGIS is applied to conduct accessibility analysis, further refining the site selection process.

**Table 2. The types, sources, and processing methods of the data used in this study.**

| Type | Data Source | Processing Process |
|---|---|---|
| POI Data | Baidu Map (https://map.baidu.com; accessed on 27 February October 2024) | Filter and deduplicate data, standardize the coordinate system, and apply spatial projection |
| AOI Data | | |
| Population Data | China's Seventh National Population Census Data (https://www.stats.gov.cn/; accessed on 27 February October 2024) | Convert into a 100m × 100m raster format in ArcGIS |
| Transportation Road Data | OpenStreetMap (https://www.openstreetmap.org/; accessed on 28 February October 2024) | Organize and classify road data in ArcGIS to generate a hierarchical road network |
| The Smart Parking Facility Data | CHINA RAILWAY 15TH BUREAU GROUP CO-RPARATION LIMITED | The data were organized and classified as detailed in S1 Table. |

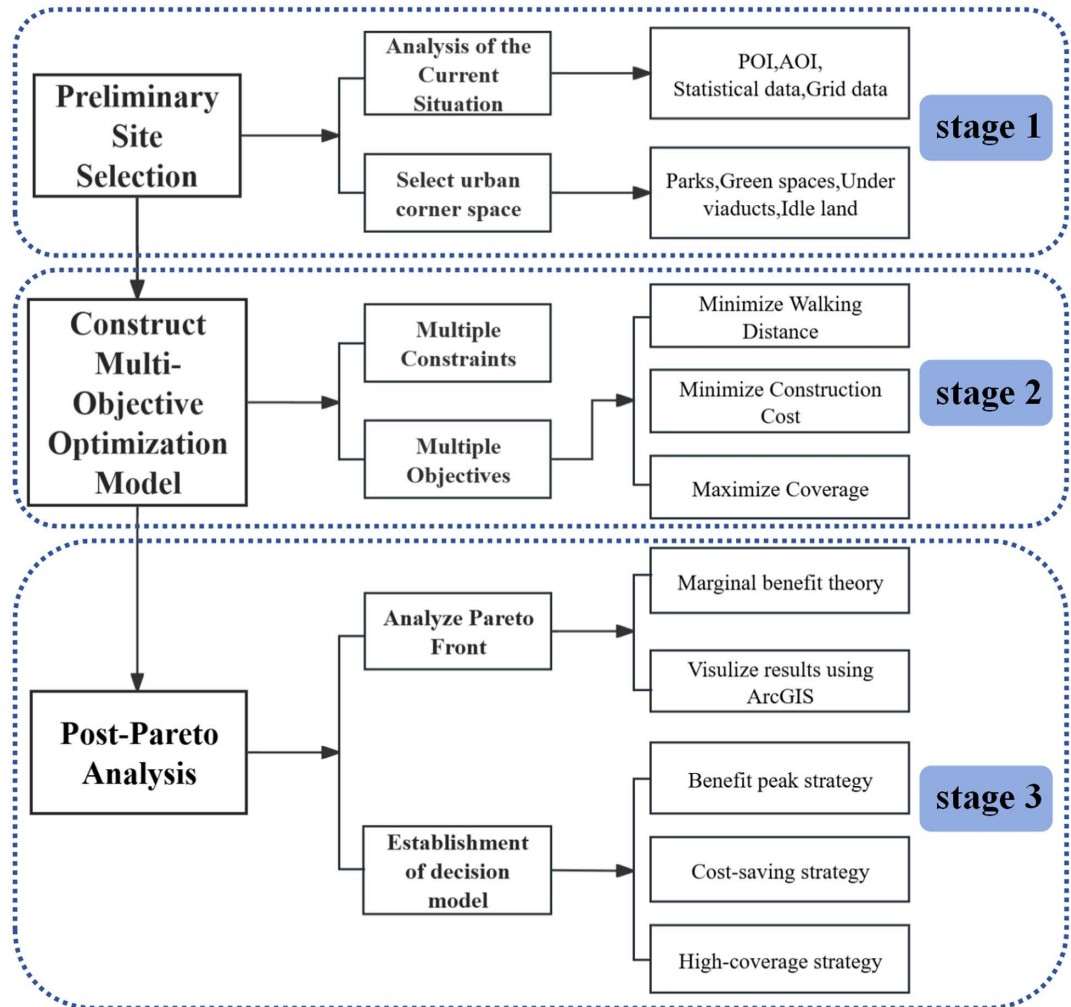

**Fig 2. Flowchart of parking lot site selection optimization research.**

(1) Kernel Density Analysis

In ArcGIS, kernel density analysis is a tool used for spatial analysis, which estimates the density of point and line features within a given area. The method involves assigning diminishing weights as the distance from the radius center increases. By calculating the contribution of different grid cells within the search radius to the center's density, a cumulative density map is generated. The formula for calculating kernel density for point features is provided in Equation (1).

$$f(x,y) = \frac{1}{nh^2} \sum_{i=1}^{n} K\left(\frac{d(x,y,x_i,y_i)}{h}\right)$$

(1)

In the formula: $f(x, y)$ represents the density value at the center of a grid cell, $n$ is the number of input points, h denotes the bandwidth or search radius, which primarily controls the smoothness of the kernel function, $d(x, y, x_i, y_i)$ indicates the distance between the input point and the grid cell center, and $K$ is the kernel function, determining the degree of influence each point has on the grid cell.

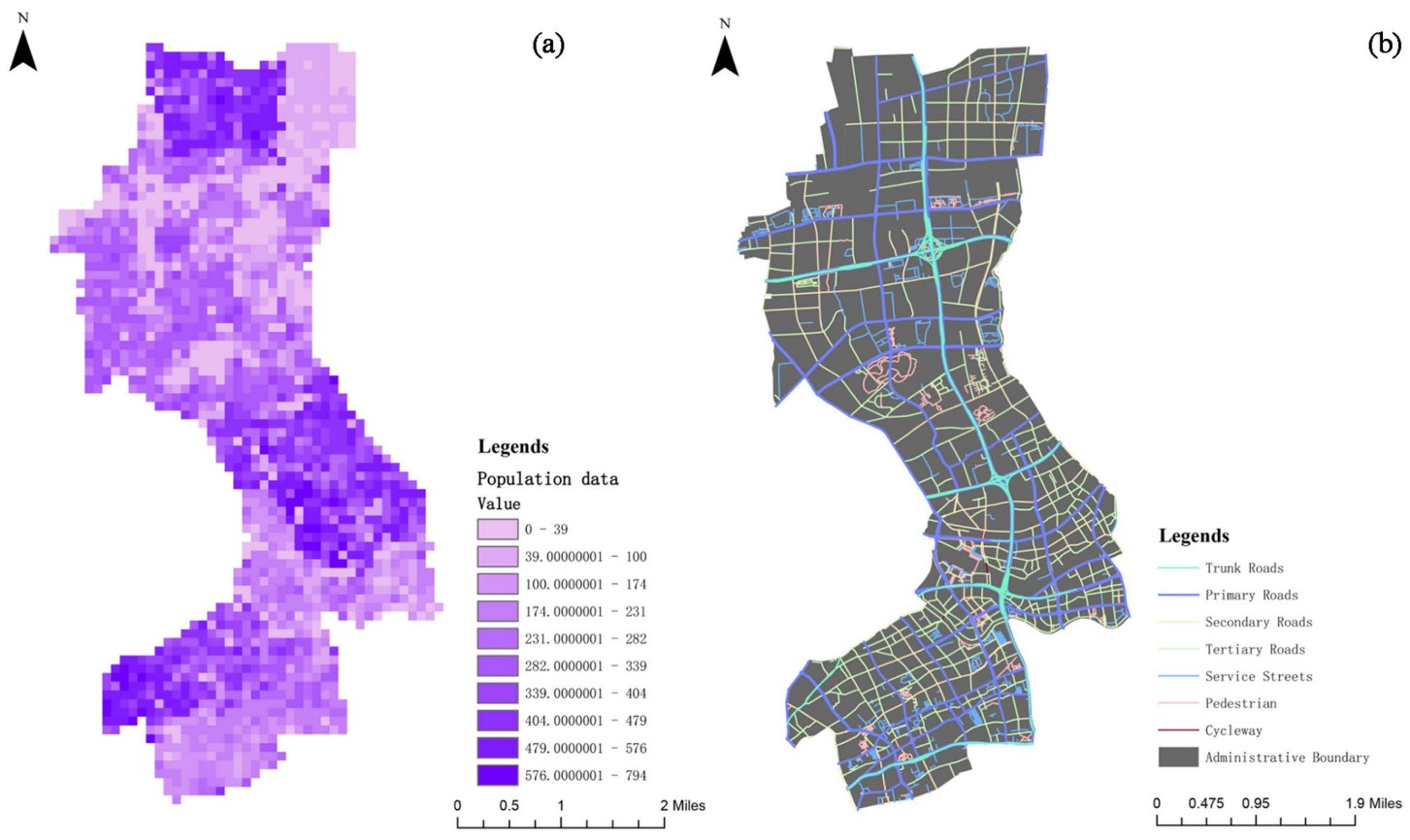

**Fig 3. Population and hierarchical extraction of road network data.**

(2) Preliminary Site Screening

To enhance land use efficiency and minimize the impact on surrounding residents, this study limits the selection of parking lot locations to urban marginal spaces. Urban marginal spaces refer to fragmented land parcels that have not been fully developed or utilized due to factors such as topography, planning constraints, or land use patterns. These parcels are typically located within urban green spaces, parks, ground spaces beneath elevated highways, urban squares, and other idle lands. Based on the above definition, and in accordance with relevant regulations (refer to Appendix- S1 Table) concerning parking lot siting requirements, suitable parking lot candidate sites were identified using ArcGIS (see Fig 4).

(3) Accessibility Analysis

Accessibility analysis is conducted using the Network Analysis Model, which is based on Graph Theory. This model constructs a road network topology consisting of nodes and edges to quantify and evaluate the spatial relationships between facility points and demand points [63]. Nodes represent intersections or endpoints of roads, while edges represent road segments, with weights assigned based on attributes such as length and travel speed. Service Area Analysis, one of the core methods within network analysis [64], is used to delineate the effective service areas of facility points. The mathematical expression for this analysis is as follows:

$$S_j = \{p_i \in P | d(p_i, f_j) \leq R\}$$

(2)

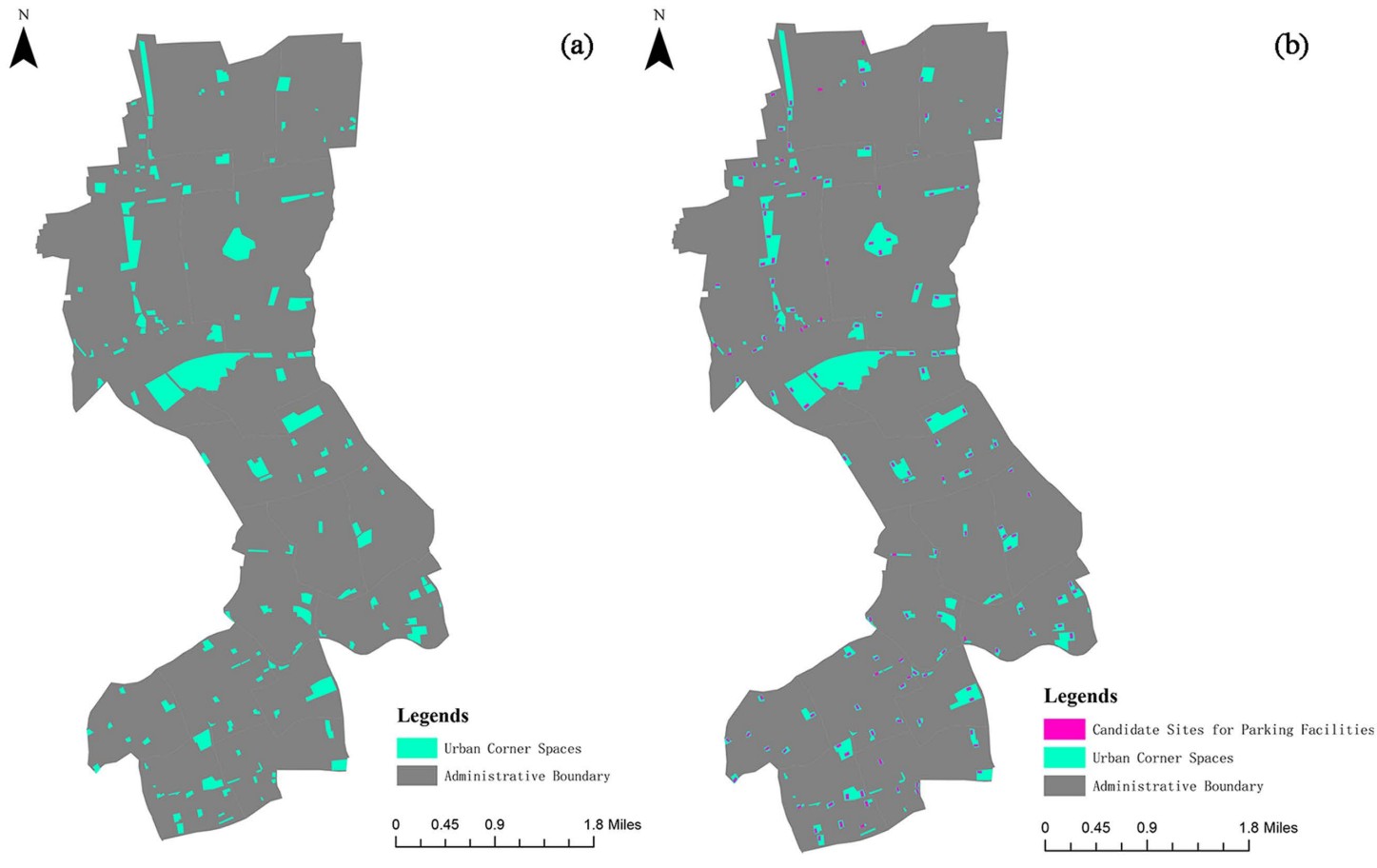

**Fig 4. Spatial distribution of urban corner spaces and selected candidate sites.**

In the formula: $S_j$ represents the service area set of parking facility $j$, $P$ denotes the set of all demand points, $p_i$ corresponds to the $i$-th demand point, $f_j$ represents the $j$-th parking facility location, $d(p_i, f_j)$ indicates the shortest path distance between demand point $i$ and parking facility $j$, and $R$ is the threshold of the service radius. The shortest path calculation is conducted using the Dijkstra algorithm, which achieves a globally optimal solution through iterative expansion of node paths.

### 3.3.2 Multi-objective optimization.

(1) Improved NSGA-II

In the multi-objective optimization phase, this study employs the Nondominated Sorting Genetic Algorithm II (NSGA-II), which effectively addresses the challenge of high computational complexity. The time complexity has been optimized from $O(MN^3)$ to $O(MN^2)$. The use of an elitist strategy in NSGA-II facilitates the preservation of superior individuals while maintaining population diversity [65]. However, NSGA-II can face challenges such as reduced search efficiency and slower convergence speed when dealing with high-dimensional problems.

To address these limitations, this study introduces a dynamic crossover and mutation probability approach, based on population crowding and evolutionary iteration numbers. This method accelerates population convergence while enriching population diversity. Moreover, the computational complexity remains unchanged, preserving an $O(MN^2)$ complexity. The specific improvements are detailed in Equations (3) and (4).

$$P_c = \begin{cases} P_{cagv} + \frac{MacIt-iter}{MacIt} \times (P_{cmax} - P_{cmin}) & , d(i) < d_{agv}(i) \\ P_{cagv} & , d(i) = d_{agv}(i) \\ P_{cagv} - \frac{iter}{MacIt} \times (P_{cmax} - P_{cmin}) & , d(i) > d_{agv}(i) \end{cases} \quad (3)$$

$$P_m = \begin{cases} P_{magv} + \frac{MacIt-iter}{MacIt} \times (P_{mmax} - P_{mmin}) & , d(i) < d_{agv}(i) \\ P_{magv} & , d(i) = d_{agv}(i) \\ P_{magv} - \frac{iter}{MacIt} \times (P_{mmax} - P_{mmin}) & , d(i) > d_{agv}(i) \end{cases} \quad (4)$$

In the formula: $P_{cagv}$, $P_{cmax}$, $P_{cmin}$, $P_{magv}$, $P_{mmax}$, $P_{mmin}$ represent the average, maximum, and minimum values of the crossover rate ($P_c$) and mutation rate ($P_m$), respectively. *iter* denotes the current iteration number, *MacIt* is the maximum number of iterations, $d(i)$ represents the crowding degree of the *i*th individual, and $d_{agv}(i)$ is the average crowding degree of the frontier to which the *i*th individual belongs.

The pseudocode for the computational process is as follows:

```
Algorithm 1. Enhanced NSGA-II with Individual Dynamic Crossover/Mutation Rates
Input: N(Population Size), MacIt, Pcagv, Pcmax, Pcmin, Pmagv, Pmmax, Pmmin, Data
Output: ParetoFront
1: P←RandomInit(N, Data) // Random Generation of Initial Populations
2: Evaluate(P, Data) // Computing Objective Function Values
3: for iter=1 to MacIt do
4:     [ranks, crowding_dist] ← NonDominatedSort(P)// Non-Dominated Sorting
5:     P←CrowdingDistanceAssignment(P, ranks, crowding_dist)
6:     d_agv←mean(crowding_dist)// Average Crowding Distance Calculation
7:     for each individual x∈P do // adaptive rates
8:         if cd(x) <d_agv then // Below Average: Gradual Perturbation Increase
9:             P_c(x) ← P_cagv + (MacIt−iter)/MacIt·(P_cmax  P_cmin)
10:            P_m(x) ← P_magv + (MacIt−iter)/MacIt·(P_mmax − P_mmin)
11:        else if cd(x)> d_agv then // Above Average: Gradual Perturbation Reduction
12:            P_c(x) ← P_cagv    iter/MacIt·(P_cmax − P_cmin)
13:            P_m(x) ← P_magv    iter/MacIt·(P_mmax − P_mmin)
14:        else              // cd=d_agv
15:            P_c(x) ← P_cagv;  P_m(x) ← P_magv
16:        end if            // Constraining Parameters Within [min, max] Range
17:            P_c(x) ← clamp(P_c(x), P_cmin, P_cmax)
18:            P_m(x) ← clamp(P_m(x), P_mmin, P_mmax)
19:    end for
20:    Q ← ∅
22:    while |Q| < N do
23:        Parent1, Parent2←TournamentSelection(P)// Binary Tournament Selection
24:        Child1, Child2←SBX_Crossover(Parent1, Parent2, P_c)
// Simulated Binary Crossover
25:        Child1←PolynomialMutation(Child1, P_m)// Polynomial Mutation
26:        Child2←PolynomialMutation(Child2, P_m)
27:        Q ← Q ∪ {Child1, Child2}
28:    end while
29:    Evaluate(Q, Data)
30:    Combined←P ∪ Q
31:    [ranks, crowding_dist] ← NonDominatedSort(Combined)
32:    P←EnvironmentalSelection(Combined, ranks, crowding_dist, N)
33: end for
34: ParetoFront ← {x∈P | x.rank=1}
35: return ParetoFront
```

(2) Algorithm effectiveness & robustness validation

To evaluate the effectiveness and robustness of the improved NSGA-II, this study incorporates benchmark testing and parameter sensitivity analysis. Specifically, to assess the performance improvements of the proposed NSGA-II, which dynamically adjusts crossover and mutation rates, four test problems were selected from the DTLZ test suite: DTLZ1, DTLZ2, DTLZ3, and DTLZ4. Table 3 outlines the configuration for each problem.

In addition to the enhanced NSGA-II introduced in this study, the original NSGA-II was also tested for comparative analysis. The parameter settings for the three algorithms were as follows: the number of iterations was set to 250, the population size was 100, and the initial crossover probability was uniformly 0.9.

Inverted Generational Distance (IGD) is widely regarded as a reliable and extensively used metric for evaluating the convergence and diversity of multi-objective evolutionary algorithms (MOEAs). Therefore, this study employs the IGD metric to assess the performance of the algorithm. The formula for calculating IGD is given as follows:

$$IGD(P, Q) = \frac{\sum_{x \in P} distance(x, Q)}{|P|}$$

(5)

In the given formula, $P$ represents a set of points uniformly distributed along the true Pareto front, indicating the number of points in $P$, while $Q$ denotes the Pareto solutions obtained from an algorithm. A lower IGD value signifies better algorithm performance.

Each algorithm was independently executed 20 times, and the average IGD values are presented in Table 4(detailed data are provided in S2 Table). Across all four test cases, it is evident that the improved NSGA-II consistently outperforms the original NSGA-II.

Furthermore, to evaluate the sensitivity of the improved NSGA-II algorithm to parameter settings, this study employs a parameter scanning and repeated experiment approach. Specifically, nine different combinations of crossover and mutation probabilities were selected, with each combination independently executed 30 times. The IGD metric obtained from each run was recorded for statistical analysis, as presented in Table 5(detailed data are provided in S3 Table).

For each parameter combination, the mean and standard deviation were calculated, and the coefficient of variation (CV) was further derived as a robustness measure. The results, summarized in Table 5, indicate that all nine CV values are below 10%, with the maximum CV being 9.3%. Consequently, it can be concluded that the algorithm's performance is not significantly sensitive to parameter variations and exhibits strong robustness.

**Table 3. Settings of DTLZ problems.**

| Problem | Number of objectives | Number of variables |
|---|---|---|
| DTLZ1 | 3 | 7 |
| DTLZ2 | 3 | 12 |
| DTLZ3 | 3 | 12 |
| DTLZ4 | 3 | 12 |

**Table 4. Average IGD values of two algorithems.**

| Problem | NSGA-II | Improved NSGA-II |
|---|---|---|
| DTLZ1 | 0.058459 | 0.048327 |
| DTLZ2 | 0.073772 | 0.065387 |
| DTLZ3 | 1.378501 | 0.993874 |
| DTLZ4 | 0.076528 | 0.072819 |

**Table 5. Robustness test results under different parameter combinations.**

| $P_c$ | $P_m$ | Average IGD | σ | CV (%) |
|---|---|---|---|---|
| 0.6 | 0.05 | 0.0629 | 0.0053 | 8.5 |
| 0.6 | 0.10 | 0.0633 | 0.0061 | 9.5 |
| 0.6 | 0.15 | 0.0683 | 0.0045 | 6.6 |
| 0.8 | 0.05 | 0.0622 | 0.0051 | 8.2 |
| 0.8 | 0.10 | 0.0659 | 0.0061 | 9.3 |
| 0.8 | 0.15 | 0.0691 | 0.0052 | 7.5 |
| 0.9 | 0.05 | 0.0604 | 0.0048 | 7.8 |
| 0.9 | 0.10 | 0.0650 | 0.0057 | 8.7 |
| 0.9 | 0.15 | 0.0692 | 0.0069 | 9.9 |

In summary, the mechanism for dynamically adjusting crossover and mutation rates effectively enhances algorithm performance while exhibiting strong robustness. This adaptability ensures that, under varying parameter settings, the algorithm consistently produces Pareto solution sets of comparable quality.

**3.3.3 Post-Pareto analysis and decision support.** In the multi-objective optimization phase, the NSGA-II algorithm is employed to generate the Pareto-optimal solution set. This study further integrates marginal benefit theory to analyze the Pareto solution set, thereby proposing a systematic and practical decision-making model. The model not only presents the Pareto-optimal solution set but also delves into the relationship between resource investment and the corresponding returns for each solution.

Marginal cost refers to the additional cost incurred by producing one more unit of a product, while marginal revenue is the additional income generated from producing one more unit. Marginal benefit is defined as the ratio of these two metrics. A higher marginal benefit indicates a higher return on investment. Diminishing marginal benefit describes the phenomenon where the incremental gains decrease as costs continue to rise [66]. The formula for marginal benefit is given as follows [67]:

$$MR = \frac{\Delta TR}{\Delta Q}$$

(6)

In the formula: $MR$ represents the marginal revenue, $\Delta TR$ denotes the change in total revenue, an $\Delta TR$ indicates the change in the quantity sold.

The proposed model assists decision-makers in identifying the most reasonable solution, specifically the critical point where marginal benefit transitions from increasing to decreasing. At this juncture, the return on investment reaches its peak, indicating that any further resource allocation would fail to generate significant incremental benefits. The analysis steps are illustrated in Fig 5.

**(1) Marginal Benefit Analysis**

A detailed marginal benefit analysis is conducted for each solution in the Pareto frontier. By fitting a curve between total construction costs and the covered population, the marginal benefit of each solution—defined as the additional population covered per unit increase in cost—is calculated. This process provides a quantitative evaluation of resource allocation efficiency underlying the Pareto-optimal solution set and clarifies the relationship between resource investment and the resulting benefits. It assists decision-makers in quantifying the actual costs and benefits of each solution, thereby identifying the most suitable options.

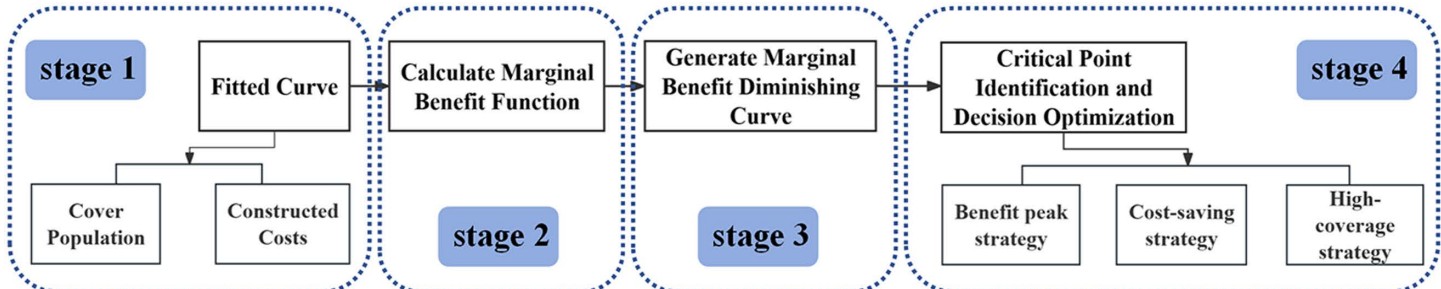

**Fig 5. Flowchart of parking lot site selection optimization research.**

(2) Critical Point Identification and Decision Optimization

Analysis of the marginal benefit curve reveals that marginal benefit increases up to a maximum point before starting to diminish. The construction cost at the critical point, denoted as $C^*$, serves as a key reference in the decision-making process, representing the optimal balance of resource allocation. By identifying this critical point where diminishing marginal benefit begins, decision-makers can determine the investment level that yields the highest return on resources.

(3) Selection of Decision Plans

Based on the analysis results, decision-makers can choose solutions with the highest marginal benefit or those closest to the critical point where diminishing returns begin as the final decision. Alternatively, solutions on the left of the critical point, characterized by lower construction costs and significant coverage improvements, may be selected in scenarios with budget constraints or during experimental phases. Conversely, solutions on the right of the critical point, which entail higher costs but broader coverage, may be prioritized in cases of higher budgets or significant parking pressures in the region. This selection process ensures optimal resource allocation and maximizes benefits across multiple objectives, meeting the diverse demands of multi-objective decision-making.

## 4. Result

### 4.1 Spatial analysis based on GIS

This study utilizes ArcGIS to perform kernel density analyses on critical areas such as residential zones, hospitals, schools, and bus stations. Fig 6(a) presents the kernel density analysis for residential zones, where high-density areas are primarily concentrated in the central and southern parts of Jing'an District. These regions, characterized by dense buildings and high population density, exhibit particularly high parking demand, especially at night.

Fig 6(b) shows the kernel density analysis for hospitals, with high-density areas mainly located in the southern part of Jing'an District, alongside notable concentrations in the northern and central regions. These areas not only accommodate significant medical service demands but also attract numerous daytime visitors, resulting in substantial parking requirements.

Fig 6(c) illustrates the kernel density analysis for schools, with high-density areas concentrated in the central and southern parts of the district. These areas host a large number of schools, along with substantial populations of students and staff. Parking demand in these regions increases sharply, particularly during peak hours associated with school arrivals and dismissals.

Fig 6(d) depicts the kernel density analysis for bus stations, with high-density areas predominantly distributed in the northeastern and southern parts of Jing'an District. These regions serve as major transportation hubs, connecting multiple residential and commercial areas. The high-density distribution of bus stations is often accompanied by elevated transfer demands. Parking facilities near these areas can significantly enhance the convenience of public transportation.

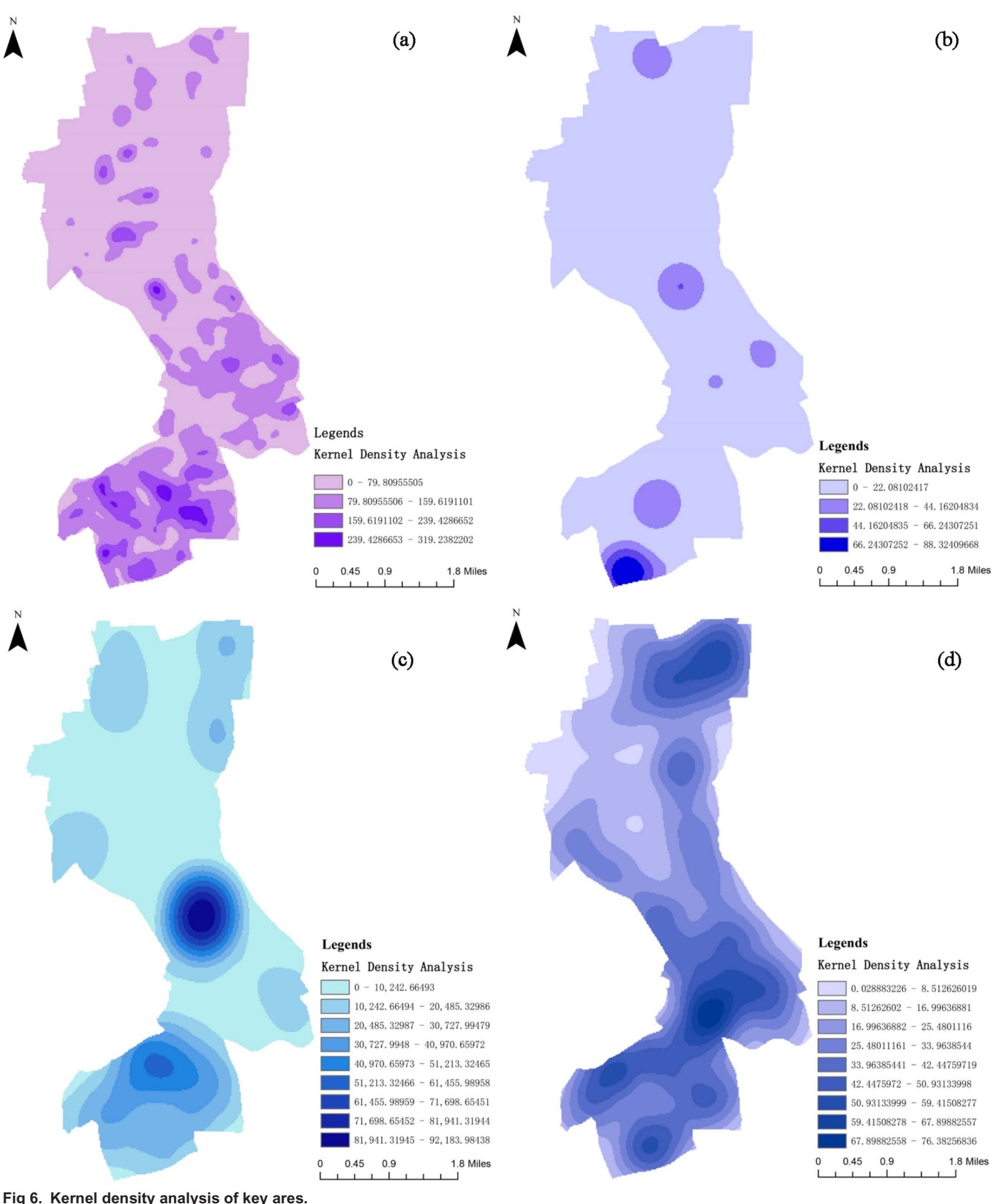

**Fig 6. Kernel density analysis of key ares.**

In summary, the northeastern, central, and southern parts of Jing'an District exhibit high parking demand. Notably, the southern region faces substantial daytime parking needs due to its medical and educational facilities, as well as elevated nighttime parking demand owing to the concentration of residential areas. Therefore, future parking lot planning should prioritize these regions.

## 4.2 Development of a multi-objective optimization model

This study establishes a multi-objective optimization model under multiple constraints, comprehensively considering the economic efficiency and service capacity of parking lot construction. The model incorporates three objective functions: maximizing population coverage, minimizing walking distance, and minimizing construction costs. The aim is to achieve optimal resource allocation among various parking demand points. Additionally, multiple constraints are established to ensure the feasibility and rationality of the solutions, including factors such as parking lot service radius, the number of parking spaces, and construction decisions.

**4.2.1 Parameter definitions.** Within the planning area, there are $n$ known parking demand points, where the coordinates of the $i$-th demand point are denoted as $(x_i, y_i)$. The parking demand at the $i$-th demand point, represented by $d_i$, is related to the population size $r_i$ within the grid containing demand point $i$. The plan includes the construction of $m$ smart parking facilities, with the coordinates of the $j$-th candidate site for parking facilities represented as $(x_j, y_j)$. The service radius of a parking facility, denoted as $R$, is set at 300 meters, which represents the effective walking distance within which the parking facility can provide services. The distance between demand point $i$ and parking facility $j$, $t_{ij}$, is calculated using the Euclidean distance formula:

$$t_{ij} = \sqrt{(x_i - x_j)^2 + (y_i - y_j)^2}$$

(7)

The binary variable $a_{ij}$ indicates whether parking facility $j$ can serve demand point $i$. If $a_{ij} = 1$, parking facility $j$ covers demand point $i$; if $a_{ij} = 0$, parking facility $j$ cannot serve demand point $i$. Similarly, the binary variable $z_j$ signifies whether a parking facility is constructed at candidate site $j$. If $z_j = 1$, a parking facility is constructed at this site; if $z_j = 0$, no facility is constructed at this site. The variable $m_j$ represents the module type of parking facility $j$, which takes an integer value ranging from 1 to 4, corresponding to four different parking facility module schemes. The construction cost of module $m_j$ is represented by $C_{mj}$, while the parking capacity of module $m_j$ is denoted by $K_{mj}$.

**4.2.2 Objective functions and constraints.** This study considers factors such as the construction costs of parking facilities, service efficiency, and user attraction to formulate three objective functions: minimizing the total walking distance, minimizing construction costs, and maximizing the number of covered users (see Table 6). At the same time, to ensure the rationality of service coverage, operational efficiency, and construction costs for parking facilities, the model must satisfy the constraints outlined in Table 7.

## 4.3 Analysis of the Pareto front solution set

The aforementioned simulation experiment was conducted using MATLAB R2024a. The algorithm parameters were configured as follows: the initial population size was set to 200, the number of iterations was set to 1,000, the initial crossover probability ($P_c$) was set to 0.8, and the initial mutation probability was set to 0.2. Based on the parameter values defined in the numerical example, the main program was executed to obtain the Pareto non-dominated solution set, as illustrated in Fig 7. This figure illustrates the interrelationships among the three objectives—minimizing construction costs, minimizing walking distances, and maximizing the number of covered users. It also highlights the distribution characteristics of location optimization schemes.

**Table 6. Optimization objectives and corresponding functions.**

| Objective | Function |
|---|---|
| Minimize walking distance | $minF_1 = \sum_{i=1}^{n} \sum_{j=1}^{m} a_{ij} \times t_{ij}$ (8) |
| Minimize construction costs | $minF_2 = \sum_{j=1}^{m} z_j \times C_{mj}$ (9) |
| Maximize the number of people reached | $maxF_3 = \sum_{i=1}^{n} \sum_{j=1}^{m} a_{ij} \times r_i$ (10) |

**Table 7. Constraints and corresponding mathematical formulations.**

| Constraints | Function |
|---|---|
| Service distance | $t_{ij} \leq R, \quad \forall i \in I, \; \forall j \in J$ |
| Candidate position | $0 \leq x_j \leq a, \; 0 \leq y_j \leq b, \quad \forall j \in J$ |
| Service capacity | $a_{ij} \leq z_j, \quad \forall i \in I, \; \forall j \in J$ |
| Coverage requirements | $z_j \leq \sum_{i=1}^{n} a_{ij}, \quad \forall i \in I, \; \forall j \in J$ |
| Binary variable | $a_{ij} \in \{0, 1\}, \; \forall i \in I, \; \forall j \in J$ |

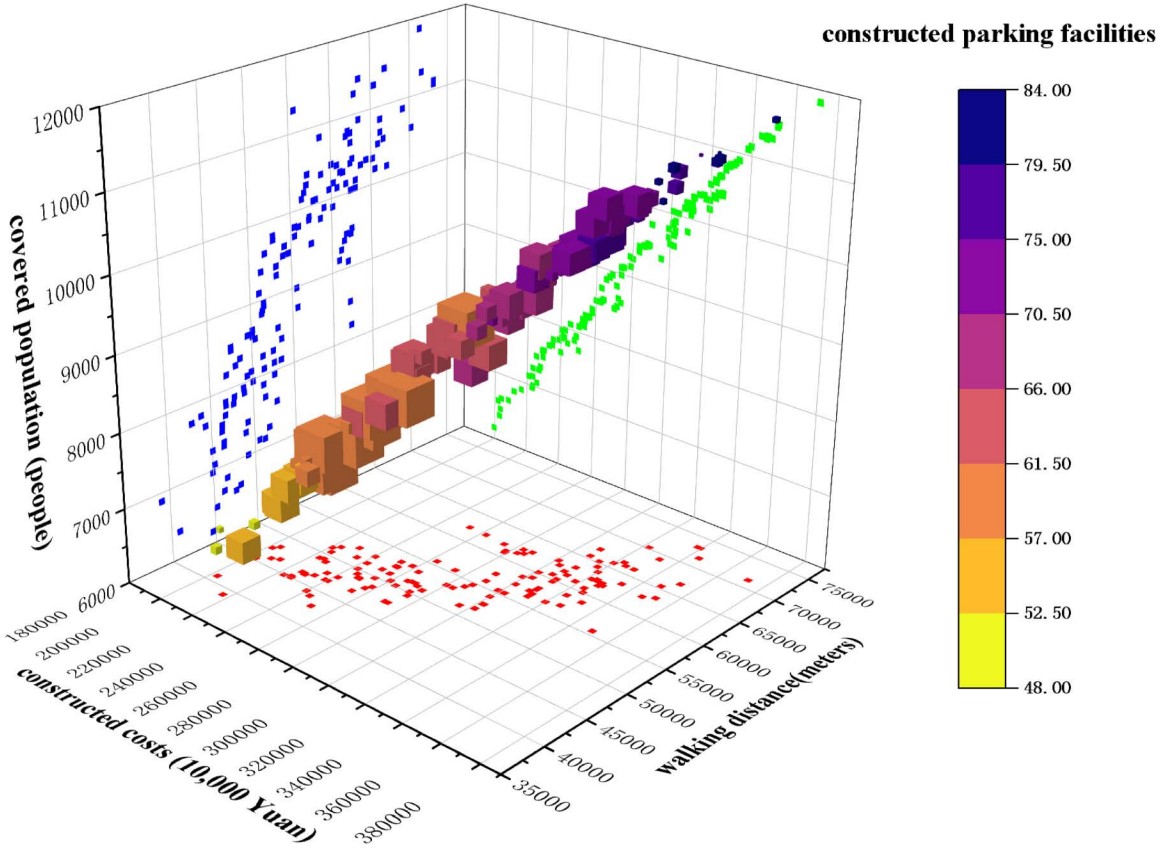

**Fig 7. 3D Pareto front.**

Firstly, regarding the relationship between the number of covered users and construction costs, the solution set exhibits a clear positive correlation. In Fig 7, the color gradient from light yellow to deep blue indicates an increase in the number of constructed parking facilities. This demonstrates that to cover more demand points, the scale and number of parking facilities must be expanded, leading to higher economic costs. However, some solutions achieve a relative increase in coverage without significantly raising costs. These solutions often represent critical points for optimization.

Secondly, the relationship between walking distance and other objectives is more complex. The figure shows that certain solutions maintain a high number of covered users within shorter walking distances, suggesting that these solutions perform well in enhancing user convenience and service efficiency. These solutions are typically associated with moderately high construction costs. Optimizing walking distance is crucial for improving parking facility accessibility, particularly in urban centers, where the walking experience directly affects the efficiency and user satisfaction of parking facilities.

The size of the points in the figure corresponds to the frequency of the number of constructed parking facilities being selected across the entire solution set. Specifically, certain construction scales are frequently chosen in multiple solutions, resulting in larger point sizes in the figure. For example, solutions involving approximately 61 parking facilities occur more frequently, and their corresponding points are larger. This suggests that during the optimization process, constructing approximately 61 parking facilities achieves a good balance across multiple objectives. Further analysis reveals that solutions with fewer facilities—fewer than 52—are often represented by smaller points and lighter colors, while solutions with more facilities—more than 80—are depicted as smaller points with darker colors. This indicates that these two types of solutions are relatively uncommon in the optimization process. The former suggests that small-scale constructions struggle to simultaneously optimize coverage, construction costs, and walking distances. The latter indicates that although these solutions achieve higher coverage, the associated cost increases make them less prevalent in the final solution set.

Further observation reveals that some solutions achieve a favorable trade-off between moderate construction costs and walking distances. These solutions are characterized by clusters of points with consistent colors, reflecting that moderate construction costs can effectively balance coverage and service range while meeting most demands. This finding provides practical guidance for decision-making: when budget constraints exist, strategically allocating the locations and scales of parking facilities can still meet most users' needs.

Overall, the Pareto non-dominated solution set highlights the complexity of multi-objective optimization and the trade-offs among different objectives. Decision-makers can select solutions that balance coverage, walking distance, and construction costs based on specific requirements as the final location plan.

## 4.4 Marginal benefit analysis and decision support

In the post-Pareto optimization phase, this study incorporates marginal benefit theory to explore the critical relationship between construction costs and the number of covered users, focusing on identifying resource allocation strategies that maximize efficiency. The results are visualized using ArcGIS, offering decision support for location optimization and providing a novel perspective for addressing multi-objective optimization problems.

In this context, marginal benefit refers to the additional number of users covered per unit increase in construction costs. To calculate marginal benefits, a fitted curve between total construction costs ($C$) and the number of covered users ($P$) from the Pareto front solution set is constructed as the basis for the analysis. The expression for this curve can be obtained using polynomial fitting, as follows:

$$P(C) = a_0 + a_1 C + a_2 C^2 + \cdots + a_n C^n \tag{11}$$

The coefficients $a_0, a_1, \ldots, a_n$ of the polynomial fitting curve are estimated using the least squares method. The resulting fitted curve, as shown in Fig 8, effectively reflects the variation trend between construction costs and the number of covered

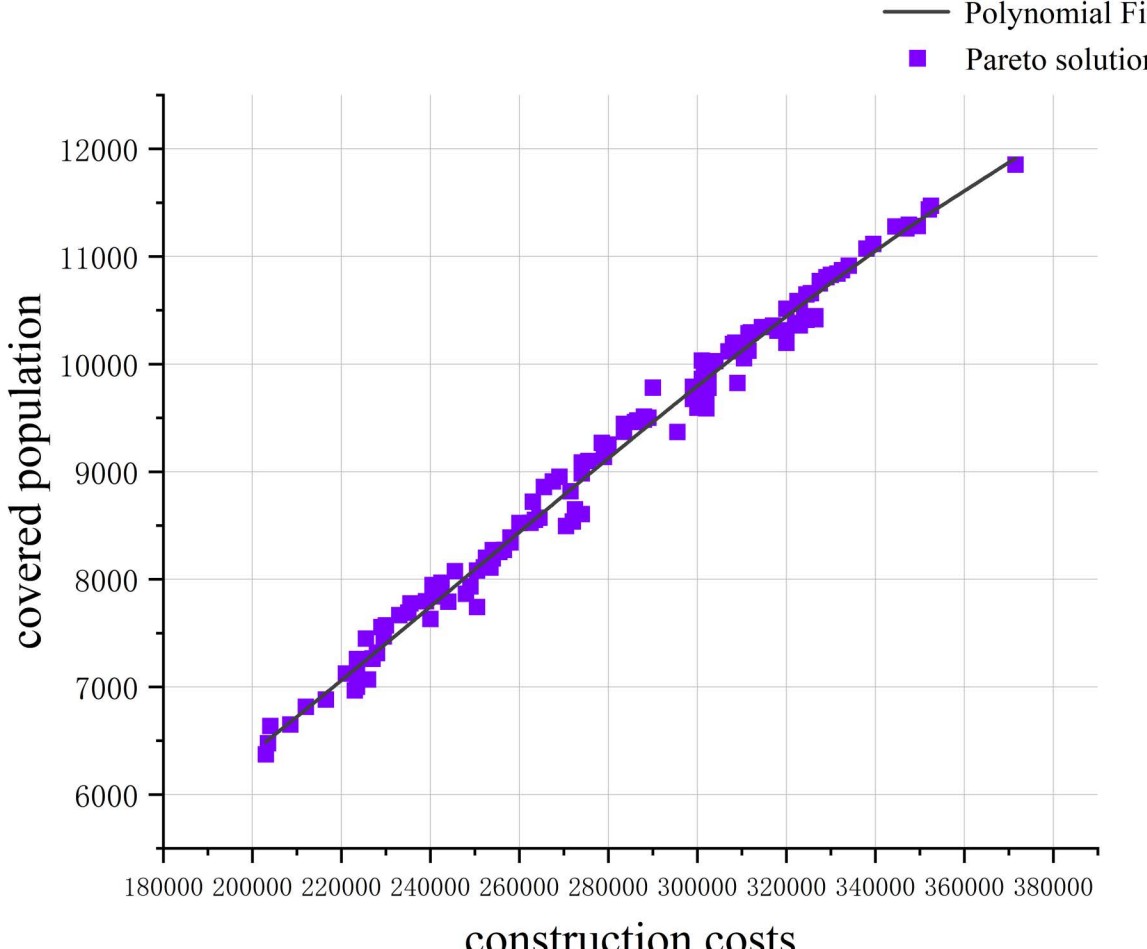

**Fig 8. Fitted curve plot.**

users. By computing the first derivative of the fitted curve $P(C)$, the marginal benefit function is derived, representing the incremental number of covered users per unit increase in construction costs:

$$MB(C) = \frac{dP(C)}{dC} = a_1 + 2a_2 C + \cdots + na_n C^{n-1} \tag{12}$$

This function represents the variation in marginal benefits under different levels of construction costs, as illustrated in Fig 9. The marginal benefit diminishing curve in the figure intuitively demonstrates the phenomenon of decreasing marginal returns as construction costs increase. Theoretically, with initial resource investment, significant gains are achieved, such as a rapid increase in the number of covered users. However, as investments continue to increase, the marginal benefits gradually diminish, meaning that each additional unit of cost contributes progressively fewer increments in the number of covered users. This phenomenon aligns with the economic principle of diminishing marginal returns, which posits that beyond a certain level of investment, the additional returns on new investments diminish over time. Such an analysis is critical for understanding the balance between resource allocation and the outcomes achieved in parking facility optimization.

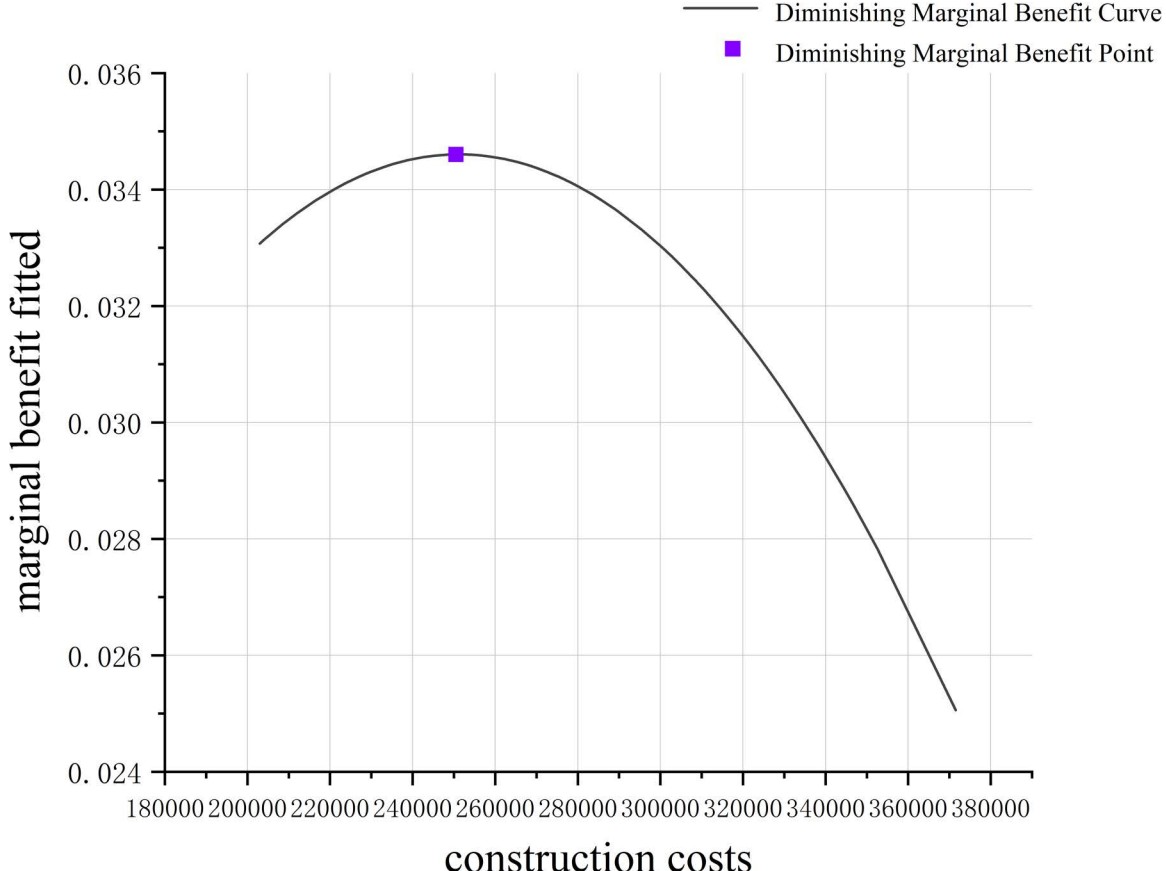

**Fig 9. Marginal benefit analysis chart.**

By further analyzing the marginal benefit curve, a peak point is identified. This point represents the construction cost and the number of covered users at the state of maximum marginal benefit. At this stage, the return on investment is highest, meaning that each unit of construction cost results in the greatest incremental increase in the number of covered users. Let the construction cost at this peak point be denoted as $C_x$. The condition for maximizing marginal benefit at this point can be expressed as:

$$\frac{d^2 P(C)}{dC^2} = 0 \tag{13}$$

By solving this condition, the critical point $C^*$ can be determined, marking the transition from increasing to decreasing marginal benefits. This point corresponds to the solution with the highest marginal rate of return. Such a solution not only enhances the service coverage capacity of parking facilities effectively but also keeps construction cost investments under control. The marginal benefit curve analysis, as described above, offers decision-makers three levels of choice guidance, as illustrated in Fig 10. These levels provide insights into optimizing resource allocation by balancing coverage, cost, and efficiency, ultimately supporting informed decision-making.

(1) Benefit Peak Strategy: For decision-makers aiming for the highest investment return, the peak point $C^*$ on the marginal benefit diminishing curve is the primary option to consider. At this point, each unit increase in cost results in the maximum additional number of covered users, making it the most economically efficient solution.

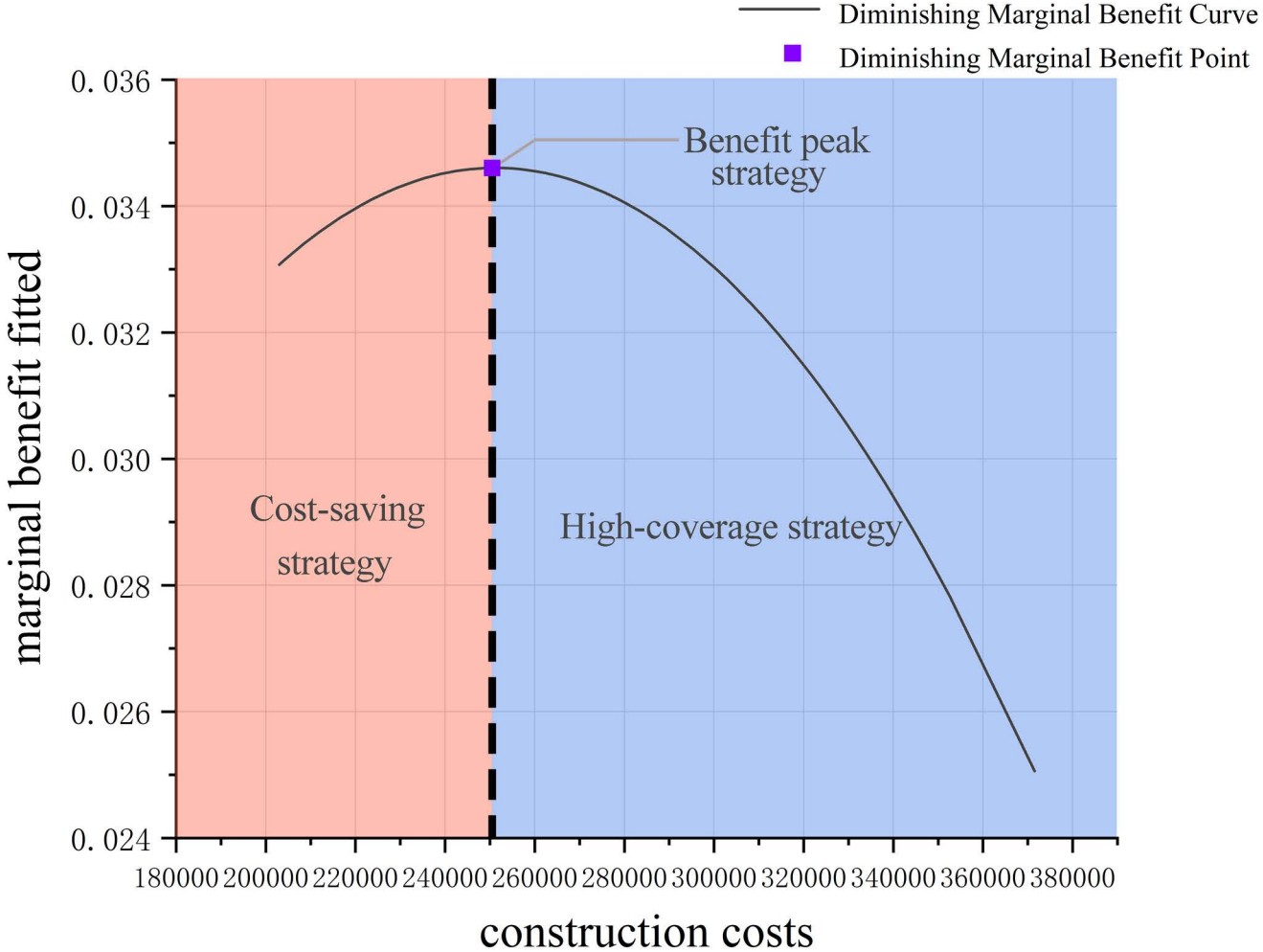

**Fig 10. Decision strategy diagram.**

(2) Cost-Saving Strategy: For decision-makers with budget constraints who aim to achieve significant coverage improvements with minimal investment, strategies within the increasing marginal benefit interval ($C < C^*$) are ideal. In this range, construction costs remain relatively low, and as investment increases, coverage improves significantly. This strategy is suited for optimizing parking facility locations under financial limitations.

(3) High-Coverage Strategy: For decision-makers who prioritize covering the maximum number of demand points, solutions in the decreasing marginal benefit interval ($C > C^*$) are recommended. In this range, although construction costs increase substantially, large-scale construction can cover more demand points, making it suitable for high-demand areas or regions with significant parking pressure.

### 4.5 Spatial analysis of the benefit peak strategy

Based on the marginal benefit analysis of the Pareto optimal solution set, the "Benefit Peak Strategy" represents the most efficient balance between investment and output. This strategy proposes adding 61 smart parking facilities in Jing'an

District, providing a total of 8,081 new parking spaces. To further evaluate the feasibility of this strategy and its impact on the regional parking structure, ArcGIS was employed to conduct spatial analysis from two perspectives:

(1) Kernel Density Analysis: A kernel density analysis of the locations of newly added parking facilities (as shown in Fig 11) reveals that the four major areas, A, B, C, and D, exhibit the highest kernel density. Referring to the study in Section 4.1 on population, hospitals, schools, and bus stops in Jing'an District, these areas currently face significant parking supply-demand gaps. The "Benefit Peak Strategy" places a relatively concentrated number of smart parking facilities in these regions, reflecting a site selection approach that integrates a "demand-oriented" focus with a "priority for key areas." By embedding parking facilities into urban "marginal spaces," such as green areas, under-bridge spaces, or vacant urban lands, the strategy not only maximizes land utilization but also avoids large-scale demolition, thus minimizing the impact on nearby residents. Moreover, this approach aligns well with the "tailored and multi-level development" strategies proposed in Jing'an District's 14th Five-Year Plan, specifically targeting parking challenges in older residential communities and around high-demand public service facilities.

(2) Pedestrian Accessibility Analysis: The pedestrian accessibility of the "Benefit Peak Strategy" was evaluated using ArcGIS network analysis (as shown in Fig 12). The results indicate that in the four key areas (A, B, C, D), the 3-minute coverage rate increased significantly, and the 7-minute reachable range exceeded 70%. For nighttime parking, the strategy effectively meets residents' commuting needs, while for daytime short-term parking, such as hospital visits or school drop-offs, it ensures reasonable decentralization and proximity to services. This layout not only reduces pedestrian walking distances but also objectively mitigates traffic congestion caused by roadside temporary parking.

Further integration with the existing public transportation network reveals that some newly constructed smart parking facilities are located close to bus stops or metro entrances, facilitating seamless intermodal travel. For residents living near metro or transit hubs, the "park-walk-ride" mode becomes more efficient, thereby alleviating road pressure in urban core areas to some extent. Based on a comprehensive assessment of current travel characteristics and demand distribution, it can be anticipated that implementing this strategy will enhance the balanced allocation and operational efficiency of parking resources.

## 5. Discussion and conclusion

### 5.1 Discussion

This study introduces an improved optimization methodology that integrates GIS-based spatial analysis with economic marginal benefit theory to construct a multi-objective location optimization framework. This approach overcomes the high dependency on weight setting in traditional Multi-Criteria Decision-Making (MCDM) methods and addresses the limitations of Multi-Objective Decision-Making (MODM) approaches, which provide Pareto solution sets but lack economic benefit evaluation. By quantifying the relationship between construction costs and population coverage, the research identifies the optimal balance of resource allocation and establishes a quantifiable decision-making model. Furthermore, the enhanced NSGA-II algorithm maintained solution diversity in the early stages and improved convergence in the later stages, avoiding local optima. Overall, this research demonstrates the potential of interdisciplinary approaches in multi-objective decision-making, providing novel insights and tools for urban planning and traffic management.

In the GIS-based spatial analysis, tools like ArcGIS were used to visualize population distribution, transportation networks, key public facilities, and marginal urban spaces in Jing'an District. The findings align with Shanghai's Implementation Opinions on Strengthening Urban Parking Facility Planning and Construction Management and the "tailored and multi-level development" strategies emphasized in the 14th Five-Year Plan. These results also corroborate urban distribution patterns derived from Point of Interest (POI) data in previous studies. For example, Li et al. [68] identified spatial agglomeration of population and jobs, along with major commuting corridors in Shanghai's urban core, based on POI

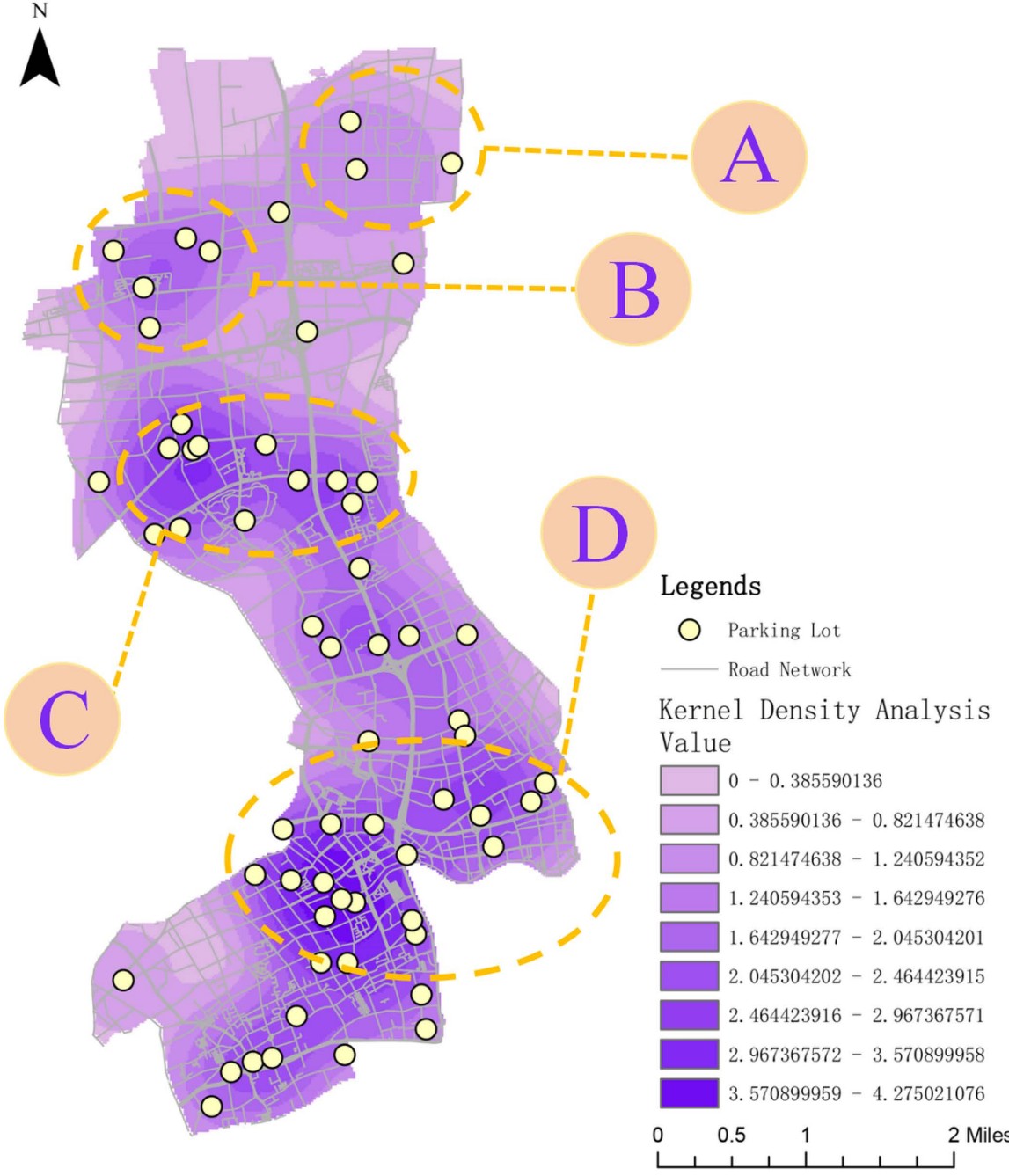

**Fig 11. Kernel density analysis of the benefit peak strategy.**

data. Their findings resonate with this study's conclusions that high-demand areas form "agglomeration corridors" within the city. Similarly, Li et al. [69] highlighted the dense and outward-expanding distribution of cultural POIs in Shanghai's urban core, which aligns with the high-density zones identified around old residential areas and high-demand public service facilities in this study. These observations highlight the capacity of GIS-based multi-source data analysis to accurately

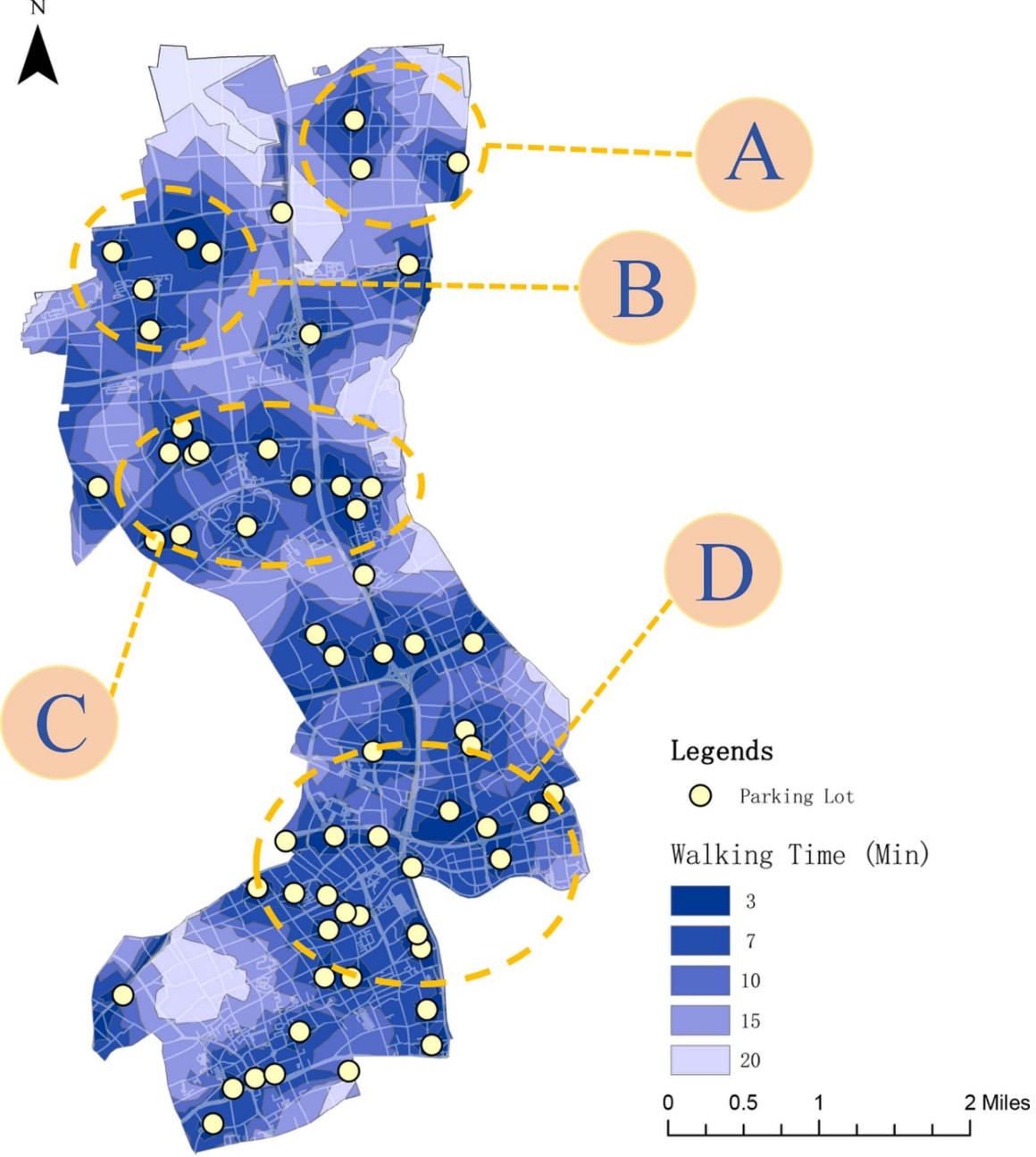

**Fig 12. Accessibility analysis of the benefit peak strategy.**

identify potential parking demand hotspots and substantiate findings with existing studies and policy guidelines, providing a solid data foundation for subsequent multi-objective optimization and differentiated decision-making.

In the context of multi-objective optimization, this study not only employs the improved NSGA-II algorithm to generate high-quality Pareto solution sets but also leverages marginal benefit theory to perform quantitative analysis of incremental benefit differences across solutions, forming a more practical decision-making model. Compared to Shen et al. [37], who

used genetic algorithms to select parking location solutions, this study dynamically adjusts crossover and mutation rates to balance solution diversity and convergence efficiency. In contrast to Hong et al. [40], who applied NSGA-II for electric vehicle charging station location but stopped at presenting Pareto solution sets, this research incorporates marginal benefit theory for a fine-grained quantitative analysis of the "construction cost-population coverage" relationship. This enables the identification of a "Benefit Peak Strategy" and provides more flexible, differentiated decision strategies. Overall, the post-Pareto analysis method constructed in this study facilitates the rapid identification of the optimal balance between resource input and return in multi-solution sets, offering new insights and transferable experience for extending multi-objective decision-making methods in urban planning and facility layout.

However, certain limitations remain. This study primarily focuses on "parking difficulties" in key areas, such as old residential neighborhoods, hospitals, and schools, aiming to maximize population coverage, minimize walking distances, and reduce construction costs. It does not differentiate parking demands across different travel types or user groups, making it difficult to achieve more refined and differentiated service strategies. Future research could classify population attributes and travel purposes to design more targeted parking facility plans for specific groups or time periods. Additionally, while the improved NSGA-II algorithm achieves a good balance of convergence and solution diversity for multi-objective optimization, its computational efficiency and stability may require further enhancement when the candidate site range expands or optimization dimensions increase. Future studies could integrate other heuristic algorithms to improve computational speed for applications in mega-cities and validate the robustness and scalability of the methodology in practical planning practices.

## 5.2 Conclusion

This paper introduces a multi-objective site selection optimization method as a scientific decision-support tool for parking lot planning. It particularly emphasizes the innovative use of urban marginal spaces, providing practical solutions for alleviating parking challenges in high-density urban areas of megacities. The key points are as follows:

(1) Algorithm Optimization

By dynamically adjusting crossover and mutation rates, the study significantly improves the traditional NSGA-II algorithm in terms of convergence efficiency and solution diversity under high-dimensional and multi-objective conditions. The improved algorithm maintains stable convergence in large-scale data scenarios, explores the solution space more evenly, and effectively avoids local optima or excessive bias toward a single objective.

(2) Spatial Analysis

In high-density urban areas, parking demand is most acute near older residential communities, hospitals, schools, and major public transit hubs. In Shanghai's Jing'an District, the southern, central, and parts of the northeastern regions face the most pronounced parking conflicts. Incorporating "urban marginal spaces" as potential parking sites not only alleviates parking pressures but also promotes the coordinated use of ecological and municipal resources, providing a more flexible and viable site selection approach.

(3) Multi-Objective Optimization

The optimization results reveal clear trade-offs: 1) Prioritizing higher population coverage leads to increased construction costs. 2) Prioritizing lower construction costs reduces walkability and service coverage. 3) Balancing construction cost, population coverage, and walkability improves urban resource efficiency, offering valuable insights for regions aiming for relatively high coverage within limited budgets.

The multi-objective optimization provides scientific guidance for various decision-making scenarios based on differing budgets and needs.

(4) Post-Pareto Analysis

The multi-strategy post-Pareto analysis offers targeted, actionable guidance for the hierarchical deployment and resource matching of smart parking facilities in high-density areas. Key strategies include: 1) Benefit Peak Strategy: Proposes constructing 61 smart parking garages to add approximately 8,081 spaces, achieving an optimal balance between cost and additional coverage. 2) Cost-Saving Strategy and High-Coverage Strategy: These cater to scenarios with constrained budgets or extensive parking demand, respectively, offering refined guidance for flexible planning under various financial and policy contexts.

This interdisciplinary approach not only has theoretical innovation but also demonstrates promising practical applications. By integrating actual urban data with detailed geographic information, this study provides a flexible and scalable decision-support framework for addressing the complex issue of urban "parking difficulties."

## Supporting information

**S1 Table. Data of the smart parking facility.**
(XLSX)

**S2 Table. IGD values under different problems.**
(XLSX)

**S3 Table. IGD values in robustness testing.**
(XLSX)

## Acknowledgments

The authors would like to thank the research team of Xiaodan Li for their valuable support and contributions to this research. Their insights and assistance have been greatly appreciated.

## Author contributions

**Conceptualization:** Xiaodan Li, Yunci Guo, Zhen Liu, Dandan Sun.

**Data curation:** Xiaodan Li, Yunci Guo.

**Formal analysis:** Xiaodan Li, Yunci Guo.

**Funding acquisition:** Xiaodan Li, Yunci Guo.

**Investigation:** Xiaodan Li, Yunci Guo, Zhen Liu.

**Methodology:** Xiaodan Li, Yunci Guo, Zhen Liu.

**Project administration:** Xiaodan Li, Yunci Guo.

**Resources:** Xiaodan Li, Yunci Guo, Dandan Sun, Wencan Wang.

**Software:** Xiaodan Li, Yunci Guo.

**Supervision:** Xiaodan Li, Yunci Guo, Zhen Liu, Dandan Sun.

**Validation:** Xiaodan Li, Yunci Guo.

**Visualization:** Xiaodan Li, Yunci Guo, Yidi Liu.

**Writing – original draft:** Xiaodan Li, Yunci Guo.

**Writing – review & editing:** Xiaodan Li, Yunci Guo, Zhen Liu.

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
