## [Decision Letter · Decision Letter 0]

PONE-D-25-14927A study on multi-objective optimization for the location selection of smart underground parking facilities in high-density urban areas of megacities: A case study of Jing'an district, ShanghaiPLOS ONE

Dear Dr. Guo,

Thank you for submitting your manuscript to PLOS ONE. After careful consideration, we feel that it has merit but does not fully meet PLOS ONE’s publication criteria as it currently stands. Therefore, we invite you to submit a revised version of the manuscript that addresses the points raised during the review process.

Contributions by artificial intelligence (AI) tools and technologies to a study or to an article’s contents must be clearly reported in a dedicated section of the Methods, or in the Acknowledgements section for article types lacking a Methods section. This section should include the name(s) of any tools used, a description of how the authors used the tool(s) and evaluated the validity of the tool’s outputs, and a clear statement of which aspects of the study, article contents, data, or supporting files were affected/generated by AI tool usage. When revising your manuscript, please consider all issues mentioned in the reviewers' comments carefully: please outline every change made in response to their comments and provide suitable rebuttals for any comments not addressed. Please note that your revised submission may need to be re-reviewed.

Authors must share the “minimal data set” for their submission. PLOS defines the minimal data set to consist of the data required to replicate all study findings reported in the article, as well as related metadata and methods. Additionally, PLOS requires that authors comply with field-specific standards for preparation, recording, and deposition of data when applicable.

We look forward to receiving your revised manuscript.

Kind regards,

Genyu Xu, Ph.D.

Academic Editor

PLOS ONE

3. We note that Figures 1,3,4,6,11 and 12 in your submission contain [map/satellite] images which may be copyrighted. All PLOS content is published under the Creative Commons Attribution License (CC BY 4.0), which means that the manuscript, images, and Supporting Information files will be freely available online, and any third party is permitted to access, download, copy, distribute, and use these materials in any way, even commercially, with proper attribution. For these reasons, we cannot publish previously copyrighted maps or satellite images created using proprietary data, such as Google software (Google Maps, Street View, and Earth). For more information, see our copyright guidelines: http://journals.plos.org/plosone/s/licenses-and-copyright.

a. You may seek permission from the original copyright holder of Figures 1,3,4,6,11 and 12 to publish the content specifically under the CC BY 4.0 license. 

4. Please remove your figures from within your manuscript file, leaving only the individual TIFF/EPS image files, uploaded separately. These will be automatically included in the reviewers’ PDF.

Reviewers' comments:

Reviewer's Responses to Questions

**Comments to the Author**

1. Is the manuscript technically sound, and do the data support the conclusions?

Reviewer #1: Yes

Reviewer #2: Partly

2. Has the statistical analysis been performed appropriately and rigorously? 

Reviewer #1: Yes

Reviewer #2: No

3. Have the authors made all data underlying the findings in their manuscript fully available?

Reviewer #1: Yes

Reviewer #2: No

4. Is the manuscript presented in an intelligible fashion and written in standard English?

Reviewer #1: Yes

Reviewer #2: Yes

5. Review Comments to the Author

Reviewer #1: This article is devoted to location selection of smart underground parking facilities. A multi-objective optimization model maximizing population coverage, minimizing construction costs, and reducing walking distances is proposed. An improved Nondominated Sorting Genetic Algorithm II (NSGA-II) is proposed. The computational results show its great performance. This paper is very interesting, after a careful review of the manuscript, I have the following comments:

1. It is suggested to list the contributions of the paper in Section 1.

2. In the related works. It is better to provide a table providing a comparison among the existing works and the proposed method on various features of the models and methods would be very handy for readers.

3. The author lacks of some state-of-the-art algorithms. It is better to compare with more state-of-the-art algorithms in terms of multi-objective evolutionary algorithms.

4. When introducing the proposed algorithm, more information such as pseudo code and computational complexity should be given.

Reviewer #2: The abstract is too long and needs to be focused and a tad shorter.

Clearly justify the selection and relevance of the three specific objectives (population coverage, construction costs, and walking distance) over other potentially relevant factors such as environmental impact or maintenance costs.

Enhance the robustness of the NSGA-II algorithm improvements by explicitly detailing the criteria and validation procedures used to set dynamic crossover and mutation rates.

Include statistical significance tests or sensitivity analyses to validate the performance improvements claimed by the dynamic NSGA-II

Refer and cite recent multi-objective optimization algorithms like MOLCA; MORKO; MaOMVO; MaOGOA, MaOWOA, etc.

Several portions of the manuscript appear as if written using LLMs, proofread and check with any AI detection tool to avoid any issues in a later stage.

6. PLOS authors have the option to publish the peer review history of their article (what does this mean? ). If published, this will include your full peer review and any attached files.

**Do you want your identity to be public for this peer review?** For information about this choice, including consent withdrawal, please see our Privacy Policy .

Reviewer #1: No

Reviewer #2: No

---

## [Author Response · Author response to Decision Letter 1]

24 May 2025

Dear Dr. Xu,

Thank you very much for providing us the opportunity to revise our manuscript. We appreciate the constructive feedback from the reviewers and the editor, which has significantly improved the quality of our paper entitled “A study on multi-objective optimization for the location selection of smart underground parking facilities in high-density urban areas of megacities: A case study of Jing'an district, Shanghai” (ID:PONE-D-25-14927). We have carefully considered and addressed all the comments provided by the reviewers and the journal editor.Changes to the manuscript are highlighted.

Below are our point-by-point responses to the editor and reviewers’ comments.

Response to the Academic Editor

We sincerely thank the Academic Editor for the valuable comments and careful review of our manuscript. We have carefully addressed the editorial concerns as follows:

1.Comment: Contributions by artificial intelligence(Al)tools and technologies to a study or to an article's contents must be clearly reported in a dedicated section of the Methods,or in the Acknowledgements section for article types lacking a Methods section.

Response: We sincerely appreciate the Academic Editor’s comments. We clarify that AI tools were not involved in any part of the research design, data processing, or result generation. The sole use of AI in our workflow was limited to English language refinement, specifically grammar and spelling correction, using the DeepSeek tool. As non-native English speakers, this helped us improve the clarity and readability of the manuscript without altering any scientific content.

2.Comment: Authors must share the "minimal data set"for their submission.

Response: To comply with the minimal data set requirement, all data necessary to replicate the findings have been provided in the supplementary files S1 Table, S2 Table, and S3 Table. These include: Data of the smart parking facility (S1), IGD values under different problems (S2), and IGD values used in robustness testing (S3). Specifically, S2 and S3 contain the raw data used to compute the means and standard deviations reported in the manuscript.

3.Comment: Please address the issue regarding image copyrights.

Response: We sincerely appreciate the editorial team’s reminder about copyright compliance. We confirm that all figures in the manuscript, including Figures 1, 3, 4, 6, 11, and 12, are either created by the authors or sourced from public domain repositories such as USGS and NSGA. Previously included data derived from Google Maps have now been entirely replaced from permitted sources.These sources permit reuse for publication under the CC BY 4.0 license. No copyrighted content from proprietary platforms like Google Maps or Earth has been used.

Response to Reviewer 1

Thank you for your insightful comments and suggestions. Please find the answers to each of your questions below

1.Comment: It is suggested to list the contributions of the paper in Section 1

Response: We sincerely appreciate the reviewer’s valuable feedback. We acknowledge that, due to an oversight and lack of refinement in our initial writing, we did not explicitly outline the contributions and innovations of this study in the introduction. In response to this, we have now incorporated a dedicated section detailing these aspects, which can be found on page 4, lines 127 to 141 of the revised manuscript.

Furthermore, to enhance readability and structural clarity, we have supplemented the introduction with a concise summary of the article’s organization. This addition is located on page 4, lines 142 to 144.

2.Comment: In the related works. It is better to provide a table providing a comparison among the existing works and the proposed method on various features of the models and methods would be very handy for readers.

Response: We sincerely appreciate the reviewer’s insightful feedback, which has been invaluable in improving the logical structure and readability of the manuscript. In response to these suggestions, we have revised the Literature Review section by incorporating a clear and intuitive comparative table. This table systematically compares several advanced multi-objective optimization algorithms alongside the algorithm proposed in this study, highlighting their respective advantages and limitations.

This addition allows readers to quickly grasp the research positioning and innovative contributions of the proposed approach. The revised content can be found on pages 6, specifically at line 203 (Table 1).

3.Comment: The author lacks of some state-of-the-art algorithms. It is better to compare with more state-of-the-art algorithms in terms of multi-objective evolutionary algorithms.

Response: We sincerely appreciate the reviewer’s valuable suggestions, which helped us recognize the need for a more comprehensive comparison of the latest advancements in multi-objective evolutionary algorithms. In response, we have supplemented the revised manuscript with an in-depth comparative analysis of several recently proposed algorithms, including but not limited to MOCLO, MORKO, and MOEDO. This addition further highlights the effectiveness and innovative aspects of the algorithm proposed in this study.

Additionally, to address the reviewer’s second concern, we have presented the algorithm comparisons in a structured table format, improving clarity and readability. These revisions can be found on page 5, lines 181 to 202 of the manuscript.

4.Comment: When introducing the proposed algorithm, more information such as pseudo code and computational complexity should be given.

Response: We sincerely appreciate the reviewer’s thorough comments. In our initial submission, we overlooked this critical aspect. Based on the reviewer’s valuable feedback, we have made the necessary revisions and additions.

In Section 3.3.2, we have now provided a more detailed description of the algorithm, including an analysis of computational complexity, pseudocode, and statistical tests assessing both algorithm effectiveness and robustness. These modifications can be found in the revised manuscript on pages 10 to 13, lines 328 to 380.

Response to Reviewer 2

We appreciate your careful review of our paper.Our answers are as follows.

1.Comment: The abstract is too long and needs to be focused and a tad shorter.

Response: We sincerely appreciate the reviewer’s valuable feedback and acknowledge that the original manuscript did not sufficiently consider this aspect. Upon further reflection, we recognize that the initial abstract was overly lengthy.

Following the reviewer’s suggestions, we have revised the abstract to be more concise and focused, emphasizing the research objectives, core methodology, and key findings. These refinements have improved clarity and readability while ensuring compliance with the journal’s formatting requirements.

The revised abstract can be found on page 2, lines 31 to 48. We are grateful for the reviewer’s thorough guidance in refining this section.

2.Comment: Clearly justify the selection and relevance of the three specific objectives ( population coverage, construction costs, and walking distance) over other potentially relevant factors such as environmental impact or maintenance costs.

Response: We sincerely appreciate the reviewer’s valuable suggestions, which highlighted the need for a more thorough explanation of the rationale behind the selected optimization objectives—population coverage, construction cost, and walking distance. In response to these comments, we have explicitly clarified the priority and relevance of these three indicators in the revised manuscript. This section has been supplemented in the introduction, specifically on page 3 to 4, lines 109 to 121.

Regarding the additional factors suggested by the reviewer, such as environmental impact and maintenance costs, we acknowledge their importance in practical projects. However, as they fall beyond the primary scope of this study, they were not the focus of our analysis. Nonetheless, we plan to incorporate a broader range of objectives in future research, adopting a more comprehensive perspective on optimization.

3.Comment: Enhance the robustness of the NSGA-ll algorithm improvements by explicitly detailing the criteria and validation procedures used to set dynamic crossover and mutation rates. Include statistical significance tests or sensitivity analyses to validate the performance improvements claimed by the dynamic NSGA-II

Response: We sincerely appreciate the reviewer’s valuable suggestions, which helped us recognize the need for a more thorough validation of algorithm effectiveness and robustness. In response to these comments, we have significantly expanded the revised manuscript to include detailed statistical significance tests and sensitivity analyses of the improved NSGA-II. These additions clearly demonstrate the substantial enhancements in convergence, diversity, and robustness of the modified algorithm compared to the original version. The relevant content has been incorporated into the manuscript and can be found on pages 11 to 13, lines 344 to 380.

4.Comment: Refer and cite recent multi-objective optimization algorithms like MOLCA; MORKO; MaOMVO; MaOGOA, MaOWOA, etc.

Response: We sincerely appreciate the reviewer’s insightful suggestions, which highlighted the need for a more comprehensive discussion of recent advancements in multi-objective optimization algorithms. In response to these recommendations, we have expanded the literature review section to include references and comparative analyses of newly proposed algorithms such as MOLCA, MORKO, MaOMVO, MaOGOA, and MaOWOA. Additionally, we have incorporated a structured table to improve readability and facilitate a clearer comparison of these methods. These revisions can be found on page 6, lines 203, Table 1 of the manuscript.

5.Comment: Several portions of the manuscript appear as if written using LLMs, proofread and check with any Al detection tool to avoid any issues in a later stage

Response: We sincerely appreciate the reviewer’s comments. It is important to emphasize that no AI was used to modify the research content, data analysis, or results during the manuscript preparation. Notably, since the authors are native Chinese speakers and English is not their first language, we exclusively utilized DeepSeek for grammar and spelling checks to ensure linguistic accuracy.

Additionally, we have thoroughly proofread the manuscript to eliminate any remaining grammatical and spelling errors. we look forward to you response regarding our submission. Please do not hesitate to contact us if there are any further questions or comments.

Sincerely,

Yunci Guo

---

## [Decision Letter · Decision Letter 1]

A study on multi-objective optimization for the location selection of smart underground parking facilities in high-density urban areas of megacities: A case study of Jing'an district, Shanghai

PONE-D-25-14927R1

Dear Dr. Guo,

We’re pleased to inform you that your manuscript has been judged scientifically suitable for publication and will be formally accepted for publication once it meets all outstanding technical requirements.

Kind regards,

Genyu Xu, Ph.D.

Academic Editor

PLOS ONE

Additional Editor Comments (optional):

Reviewers' comments:

Reviewer's Responses to Questions

**Comments to the Author**

1. If the authors have adequately addressed your comments raised in a previous round of review and you feel that this manuscript is now acceptable for publication, you may indicate that here to bypass the “Comments to the Author” section, enter your conflict of interest statement in the “Confidential to Editor” section, and submit your "Accept" recommendation.

Reviewer #1: (No Response)

Reviewer #2: (No Response)

2. Is the manuscript technically sound, and do the data support the conclusions?

Reviewer #1: (No Response)

Reviewer #2: (No Response)

3. Has the statistical analysis been performed appropriately and rigorously? 

Reviewer #1: (No Response)

Reviewer #2: (No Response)

4. Have the authors made all data underlying the findings in their manuscript fully available?

Reviewer #1: (No Response)

Reviewer #2: (No Response)

5. Is the manuscript presented in an intelligible fashion and written in standard English?

Reviewer #1: (No Response)

Reviewer #2: (No Response)

6. Review Comments to the Author

Reviewer #1: The authors have well revised this paper according to the suggestions, and I would like to recommend this manuscript for publication.

Reviewer #2: (No Response)

7. PLOS authors have the option to publish the peer review history of their article (what does this mean? ). If published, this will include your full peer review and any attached files.

**Do you want your identity to be public for this peer review?** For information about this choice, including consent withdrawal, please see our Privacy Policy .

Reviewer #1: No

Reviewer #2: No

---

## [Editor Report · Acceptance letter]

PONE-D-25-14927R1

PLOS ONE

Dear Dr. Guo,

I'm pleased to inform you that your manuscript has been deemed suitable for publication in PLOS ONE. Congratulations! Your manuscript is now being handed over to our production team.

Kind regards,

on behalf of

Dr. Genyu Xu

Academic Editor

PLOS ONE